# Moment Constrained Optimal Transport for Control Applications

**Thomas Le Corre** *thomas.le-corre@inria.fr Inria, Paris, France and Département d'informatique de l'ENS, ENS, CNRS, PSL University, Paris, France*

**Ana Bušić** *Inria, Paris, France and Département d'informatique de l'ENS, ENS, CNRS, PSL University, Paris, France*

**Sean Meyn** *Department of ECE at the University of Florida, Gainesville*

**Reviewed on OpenReview:** *https://openreview.net/forum?id=2hAtSpnat9*

## Abstract

This paper concerns the application of techniques from optimal transport (OT) to mean field control (MFC), in which the probability measures of interest in OT correspond to empirical distributions associated with a large collection of controlled agents. The control objective of interest motivates a one-sided relaxation of OT, in which the first marginal is fixed and the second marginal is constrained to a "moment class": a set of probability measures defined by generalized moment constraints. This relaxation is particularly interesting for control problems as it enables the coordination of agents without the need to know the desired distribution beforehand. The inclusion of an entropic regularizer is motivated by both computational considerations, and also to impose hard constraints on agent behavior. A computational approach inspired by the Sinkhorn algorithm is proposed to solve this problem. This new approach to distributed control is illustrated with an application of charging a fleet of electric vehicles while satisfying grid constraints. An online version is proposed and applied in a case study on the ElaadNL dataset containing 10,000 electric vehicle charging sessions in the Netherlands. This empirical validation demonstrates the applicability of the proposed approach to optimizing flexibility while respecting grid constraints.

## 1 Introduction

We address the following control problem (1), motivated by applications involving the coordination of a large population of electricity-consuming agents subject to global constraints, such as a maximum aggregate power consumption.

Consider a set of $K$ homogeneous agents, each characterized by a state:

$$X_k = (S_k, W_k) \in \mathcal{X}, \quad 1 \le k \le K.$$

Here, $S_k$ denotes a non-controllable variable, while $W_k$ is a control (or decision) variable. An electric vehicle (EV) charging use case, providing a detailed concrete example, is presented in Section 3.

We consider a central planner that coordinates a large population of agents to minimize a cost $c\colon \mathcal{X} \to \mathbb{R}$, subject to $M$ aggregate constraints on the total population (e.g. requiring that the total power consumption of all agents remain below a prescribed threshold during $M$ time steps: $\forall m \in \{1, \ldots, M\}$, $\sum_{k=1}^{K} f_m(X_k) \le 0$, where $f\colon \mathcal{X} \to \mathbb{R}^M$ denotes a mapping from state $X$ to a consumption profile $f(X)$ of an agent, and $f_m(x)$ is the m-th component of $f(x)$. The optimization problem for the central planner can thus be expressed as

$$\min_{W_k} \left\{ \sum_{k=1}^{K} c(X_k) \ : \ \sum_{k=1}^{K} f(X_k) \le 0 \right\}, \tag{1}$$

where the inequality is on each component $m \in \{1, \dots, M\}$. The use case considered in Sections 3 and 4 is the control of a large population of EVs whose state $S_k$ is the EV's arrival time at the parking lot and its initial state of charge, and the control $W_k$ is the starting charging time. The cost $c$ in this case could be seen as a penalty for deviating from a planned schedule.

A major challenge with this class of problems is that the computational complexity grows with the number of agents. For instance, the dataset considered in Section 4 contains several thousand EV charging sessions. Individually coordinating each agent while satisfying its own specific operational constraints is therefore often computationally prohibitive in practice, or may result in suboptimal choices. A popular approach to address this scalability issue is to adopt an MFC framework. In this setting, the number of agents is assumed to be sufficiently large so that the impact of any single agent on the aggregate variables becomes negligible (Lasry & Lions, 2007). What matters is then the control of the entire population, modeled through a probability distribution $\mu$ of $X = (S, W)$ over $\mathcal{X}$, and we denote $\nu$, the distribution of the non-controllable variable $S$. This gives us the mean field problem approximating equation 1 when $K$ is large:

$$\min_{\mu}\Big\{\int_{\mathcal{X}} c(x)d\mu(x) \ : \ \forall s \in \mathcal{S} \int_{\mathcal{W}} d\mu(s, w) = \nu(s) \ \text{ and } \ \int_{\mathcal{X}} f(x)d\mu(x) \le 0\Big\}. \tag{2}$$

It is important to note that the optimization is only done on the control variable $W$ and the distribution $\nu$ is not modified; this is what we will subsequently call "preserving the distribution of the non-controllable variables".

Looking at equation 2, one may interpret the problem as transporting the initial probability distribution of $X$ to another distribution, as in the OT theory. The key difference here is that the target distribution is not known a priori. In fact, determining this distribution is precisely the objective of the problem (e.g. the optimal charging policy for the fleet of EVs). The only available information about this target distribution comes from the aggregate constraints $\int_{\mathcal{X}} f(x)\,d\mu(x) \le 0$, which will hereafter be referred to as moment constraints. Recent works have introduced the Moment-Constrained Optimal Transport (MCOT) framework (Alfonsi et al., 2020), in which both the initial and final distributions are required to satisfy moment constraints, with the aim of approximating the classical OT problem, under appropriate assumptions, when the number of moment constraints tends to infinity. The novelty of our work is that we propose a new approach for modeling MFC problems as a one-sided MCOT variant, in which only the final distribution is required to satisfy the moment constraints. This one-sided variant is particularly well-suited for MFC, where the initial distribution corresponds to the nominal dynamics of the system, while the target one is only specified through the moment constraints. For example, in the demand response applications in power grids, the coordinator only cares about the constraints on the power consumption of the whole population of flexible devices, and not about the detailed distribution over all individual device state trajectories.

**Contributions** Our contributions are the following:

- We propose a new problem *Moment Constrained Optimal Transport for Control* (MCOT-C) inspired by OT and designed to achieve MFC goals: (i) Agents are controlled to meet a global constraint; (ii) Their individual hard constraints must be satisfied, either physical (e.g. an EV cannot be plugged in before it arrives, and its state of charge on arrival, or its departing time cannot be controlled) or in terms of quality of service (e.g. each EV must be fully charged when leaving). A tunable regularizing term, similar to the one introduced in entropic OT, is introduced for computational reasons but also to achieve the goal (ii).

- We propose a projected gradient descent algorithm to solve MCOT-C and highlight its similarity to the Sinkhorn algorithm.

- We extend this approach to an online setting, where the data about the EVs are progressively discovered and show its applicability on a case study with a real data set (OpenDataset, 2019).

- Compared to the existing literature on MFC for demand response, our model allows to take into account broader set of global constraints (e.g. aggregate power consumption ramping rate).

**Literature** Many academic communities are interested in efficiently transforming probability measures. Examples include the fully probabilistic control design of Kárný (1996) and the related linearly-solvable Markov decision framework (Todorov, 2007). Several methods have been designed in the field of MFC or ensemble control, with applications ranging from power systems to medicine (Hochberg et al., 2006; Chertkov & Chernyak, 2018). These techniques can also be relaxed (Cammardella et al., 2020; Bušić & Meyn, 2018) or regularized, often via a Kullback-Leibler term (Chertkov & Chernyak, 2018; Todorov, 2007) for computational reasons. Similar objectives of controlling a large population of electricity-consuming agents have been explored in the distributed control framework (Chertkov & Chernyak, 2018; Cammardella et al., 2020; Bušić & Meyn, 2018). In some cases, the problem is overly constrained by the global constraints, making it difficult or even impossible to identify feasible solutions. To address this, relaxations of the functions $f$ via quadratic penalties have been proposed (Cammardella et al., 2020; Bušić & Meyn, 2018). Additional examples and surveys can be found in Garrabe & Russo (2022).

OT theory first emerged in the 18th century, and more recently has become a significant tool in the machine learning toolbox (Villani, 2008; Peyré et al., 2019). The goal is simply described: given two random variables $X$ and $Y$, find a joint probability measure $\pi^*$ for the pair $(X, Y)$ that preserves the marginals, and minimizes a given cost. The introduction of an entropic regularizer, which leads to solutions that are easily computable by the Sinkhorn algorithm (Cuturi, 2013), has become standard in OT. This development led to the entropic optimal transport problem, which is closely related to the one considered here (except that a moment constraint replaces the constraint on the second marginal), both in its formulation and in the algorithms used to solve it. Several authors have proposed relaxations on the marginals of the OT problem, such as unbalanced OT, where an entropic penalization of the deviation from the marginals is introduced (Chizat, 2017). Relaxations of marginals have been considered to improve numerical performance or to approximate the OT problem (Balaji et al., 2020; Le et al., 2021; Alfonsi et al., 2020) but, to the best of our knowledge, never as a natural representation of an MFC problem.

Connections between OT and control theory have been well established, most notably through the Benamou–Brenier formulation (Benamou & Brenier, 2000), which bridges OT and fluid mechanics. From a control-theoretic viewpoint, this formulation can be interpreted as an optimal control problem in which an initial distribution (at the beginning of a time horizon) is transported toward a target distribution (at the end of the horizon), while minimizing the cumulative cost incurred along the trajectory. More recent research has further strengthened this link by connecting OT with dynamic programming and multi-marginal formulations (Terpin et al., 2024), thereby showing that certain classes of optimal control problems can be recast as OT problems. Another approach (Liu et al., 2022) addresses a mean field game where the target distribution is known exactly and must be reached, using Deep Reinforcement Learning. In the present work, we do not adopt the Benamou–Brenier framework; instead, the distributions considered here should be understood as policies over the course of a day, rather than states being transported.

**Notation** The state space $\mathcal{X}$ is assumed to be a closed subset of $\mathbb{R}^N$ with $N \geq 1$ and we denote $\mathcal{B}(A)$ the set of Borel probability measures on a given set $A$. For $\pi$ a bivariate distribution on $\mathcal{X}$, its marginals will be denoted $\pi_1$ and $\pi_2$ such that $\forall x \in \mathcal{X}, \pi_1(x) = \int_{\mathcal{X}} \pi(x, dy)$ and $\forall y \in \mathcal{X}, \pi_2(y) = \int_{\mathcal{X}} \pi(dx, y)$. We denote $\mathcal{U}(\mu_1, \mu_2) = \{\pi \in \mathcal{B}(\mathcal{X} \times \mathcal{X}) : \pi_1 = \mu_1, \ \pi_2 = \mu_2\}$, the set $\mathcal{B}(\mathcal{X} \times \mathcal{X})$, the set of bivariate distributions having for marginals $\mu_1$ and $\mu_2$. The scalar product for bivariate distributions $\pi$ and measurables functions $g$ is defined by $\langle \pi, f \rangle := \int_{\mathcal{X} \times \mathcal{X}} f(x, y) \pi(dx, dy)$.

## 2 Moment constrained optimal transport for control

### 2.1 Statement of the problem

The goal of the OT problem introduced by Kantorovich (1942) is to find a transport plan $\pi$ minimizing a cost $\langle \pi, c \rangle = \int_{\mathcal{X} \times \mathcal{X}} c(x, y) d\pi(x, y)$ subject to the constraint that the marginals of $\pi$ are exactly $\mu_1$ and $\mu_2$. In this section, we aim to formulate the MFC problem introduced in equation 2, in the framework of OT, where the second marginal is not fully specified but must belong to a set of distributions respecting moment constraints.

In this framework, the first marginal $\pi_1$ corresponds to a nominal behavior $\mu_1$, which is given, whereas the second marginal, denoted by $\pi_2$, corresponds to the optimized distribution within the following *moment class*,

$$\mathcal{P}_f = \{\mu \in \mathcal{B}(\mathcal{X}) : \langle \mu, f_m \rangle \leq 0 \ \forall \ 1 \leq m \leq M\}, \tag{3}$$

where $f \colon \mathcal{X} \to \mathbb{R}^M$ encodes $M$ constraints. An equality constraint $\langle \mu, f_m \rangle = 0$ can be expressed as a pair of inequality constraints, thus equality constraints can also be imposed when required. Recall that in MFC, the distribution $\nu$ of the non-controllable variable is fixed. In this framework, this implies that the bivariate distribution $\pi$ belongs to

$$K(\mu_1) = \{\pi \in \mathcal{B}(\mathcal{X} \times \mathcal{X}) : \pi((x_s, x_w), (y_s, y_w)) = \mu_1(dx_s, dx_w)T((x_s, x_w), dy_w)\delta_{x_s}(dy_s)\},$$

where $\delta$ is the Dirac measure, and $T$ ranges over all probability kernels. That is, if $\pi \in K(\mu_1)$, then $\int_{\mathcal{W}} \pi_2(y_s, dy_w) = \int_{\mathcal{W}} \pi_1(y_s, dx_w) = \nu(y_s)$, which corresponds to our goal of preserving $\nu$ on $\mathcal{S}$, for both marginals. Lastly, we will use the following Kullback-Leibler (KL) regularizer, similar to that in Cuturi (2013):

$$D_{\mathrm{KL}}(\pi \| \mu_1 \otimes \mu_2) = \int_{\mathcal{X} \times \mathcal{X}} \log\left(\frac{\pi(x, y)}{\mu_1(x)\mu_2(y)}\right) \pi(dx, dy). \tag{4}$$

However, in our case, $\mu_2$ is not the second marginal of $\pi$, but left as a design parameter (in our case the second marginal is not known a priori, as it is only constrained to belong to $\mathcal{P}_f$). This regularizer is introduced for three reasons. First, the minimization of a KL-divergence enforces absolute continuity between $\mu_2$ and $\pi_2$, meaning that the support of $\pi_2$ is included in the support of $\mu_2$. In practice, one defines the support of $\mu_2$ as the set of physically feasible or desirable states and controls (e.g., ensuring that electric vehicles are charged at the end of their time slot or that charging does not occur before arrival), thereby preventing $\pi_2$ from assigning mass to physically impossible or undesirable states. Second, unlike in certain MFC methods where the entropic penalization is taken with respect to $\mu_1$ (Bušić & Meyn, 2018), here one has the flexibility to design $\mu_2$. For instance, one may rather rely on a heuristically designed $\mu_2$ that may give faster convergence, or, in the absence of one, choose a uniform distribution over the previously defined support (that may be different than the one of $\mu_1$). Finally, it has a computational interest, as it allows for obtaining explicit solutions, as shown in Section 2.

This allows us to introduce the MFC problem:

**Problem MCOT-C:** *Moment Constrained Optimal Transport for Control*

$$\min_{\pi}\big\{\langle \pi, c \rangle + \varepsilon D_{\mathrm{KL}}(\pi \| \mu_1 \otimes \mu_2) : \pi \in K(\mu_1), \ \pi_2 \in \mathcal{P}_f\big\}. \tag{5}$$

## 2.2 Dual problem

This subsection defines the dual and the theoretical properties needed for the algorithm. More details on duality theory and proofs may be found in the appendices A and B. The theoretical results of this problem in the Gaussian case are presented in appendix C. An example that illustrates the impact of regularization can be found in appendix D.

**Assumptions** Throughout this work, we will consider the following assumptions:

**(A1)** $c \colon \mathcal{X} \times \mathcal{X} \to \mathbb{R}_+$ and $f \colon \mathcal{X} \to \mathbb{R}^M$ are continuous, and there is an open neighborhood $N \subset \mathbb{R}^M$ containing 0 such that $\mathcal{P}_{f-r}$ is non-empty for all $r \in N$. The latter condition implies a robust feasibility under small perturbations, providing a Slater-type condition necessary for strong duality.

**(A2)** $\mu_1$ and $\mu_2$ have compact support, and the problem is feasible under perturbations: for any $r \in N$, there is $\pi$ satisfying $\pi_2 \in \mathcal{P}_{f-r}$ and $\pi_1 = \mu_1$. This guarantees the existence of feasible transport plans.

**(A3)** $\Sigma^0 := \mathrm{Cov}(Y)$ is positive definite when $Y \sim \mu_2$. This implies non-redundancy of constraints, strict convexity of the dual problem and uniqueness of the Lagrange multiplier $\lambda^*$. From a numerical point of view, it provides numerical stability for gradient descent algorithms.

In the EV charging use case in Section 3, the cost $c(x, y)$ (quadratic penalties on charging time deviations) and the aggregate constraint functions $f(m)$ (bounds on the maximum power consumption or its gradient) are continuous, and distributions $\mu_1$ and $\mu_2$ have bounded support (defined by intervals). Assumption (A3) is, in particular, not true if $Y \sim \mu_2$ has linear dependence. In a concrete setting, (A3) implies that each variable in $Y$ is not a linear combination of the others. If this is the case, it is easy to change $Y$ by removing this variable.

**Dual** The dual of MCOT-C is by definition the function $\varphi^* \colon \mathbb{R}_+^M \to \mathbb{R} \cup \{-\infty\}$,

$$\varphi^*(\lambda) = \varepsilon \min_\pi \left\{ -\varepsilon^{-1} \langle \pi, \ell_0^\lambda \rangle + D_{\mathrm{KL}}(\pi \| \mu_1 \otimes \mu_2) : \pi \in K(\mu_1) \right\}, \tag{6}$$

where we introduce the notation $\ell_0^\lambda(x, y) = -\lambda^\intercal f(y) - c(x, y), \quad \forall x, y \in \mathcal{X}$.

For each $\lambda \in \mathbb{R}_+^M$, $\varepsilon > 0$ and $x = (x_s, x_w) \in \mathcal{X}$, we denote

$$B_{\lambda, \varepsilon}(x) = \varepsilon \log \int_{y_w \in \mathcal{W}} \exp\left( \varepsilon^{-1} \ell_0^\lambda((x_s, x_w), (x_s, y_w)) \right) \mu_2(dy_w). \tag{7}$$

**Proposition 1.** *Subject to (A1)–(A3),*

**(i)** *The infimum equation 6 gives $\varphi^*(\lambda) = -\langle \mu_1, B_{\lambda, \varepsilon} \rangle$.*

**(ii)** *The maximizer is $\pi^\lambda(x, y) = T^\lambda(x, y)\mu_1(x)$ with $\forall x = (x_s, y_s) \in \mathcal{X}, \forall y = (x_s, y_s) \in \mathcal{X}$,*

$$T^\lambda(x, y) = \mu_2(y)\delta_{x_s}(y_s) \exp(L^\lambda(x, y)), \qquad L^\lambda(x, y) = \varepsilon^{-1}\{\ell_0^\lambda(x, y) - B_{\lambda, \varepsilon}(x)\}, \tag{8a}$$

*and $\mu^\lambda(y) = \pi_2^\lambda(y) \quad \forall y \in \mathcal{X}$.*

**(iii)** *There is no duality gap: there is a unique $\lambda^* \in \mathbb{R}_+^M$ satisfying*

$$\varphi^*(\lambda^*) = \min_\pi \left\{ \langle \pi, c \rangle + \varepsilon D_{KL}(\pi \| \mu_1 \otimes \mu_2) : \pi \in K(\mu_1), \ \pi_2 \in \mathcal{P}_f \right\}. \tag{8b}$$

It is convenient to make the change of variables $\zeta = \varepsilon^{-1}\lambda$, and consider $\mathcal{J}(\zeta) := -\varepsilon^{-1}\varphi^*(\varepsilon\zeta)$.

We turn next to the representation of the derivatives of the dual function. The quantity $\varepsilon^{-1}B_{\varepsilon\zeta, \varepsilon}(x)$ is a log moment generating function for each $x$; for this reason, it is not difficult to obtain suggestive expressions for the first and second derivatives with respect to $\zeta$.

**Proposition 2.** *The function $\mathcal{J}$ is convex and continuously differentiable. The first and second derivatives of $\mathcal{J}$ admit the following representations:*

$$\nabla \mathcal{J}(\zeta) = m^\lambda, \qquad \nabla^2 \mathcal{J}(\zeta) = \Sigma^\lambda, \tag{9a}$$

*in which $m_i^\lambda = \langle \mu^\lambda, f_i \rangle = \mathbb{E}^\lambda[f_i(Y)]$ for each $i$, and the Hessian equation 9a coincides with the conditional covariance:*

$$\Sigma^\lambda = \mathbb{E}^\lambda[f(Y)f(Y)^T] - \mathbb{E}^\lambda\big[\mathbb{E}^\lambda[f(Y) \mid X]\mathbb{E}^\lambda[f(Y) \mid X]^T\big]. \tag{9b}$$

It follows that $\mathcal{J}$ is strictly convex.

**Lemma 1.** *Suppose that (A1)–(A3) hold. Then, the covariance $\Sigma^\lambda$ is full rank for any $\lambda \in \mathbb{R}_+^M$.*

### 2.3 Algorithm: Semi-Sinkhorn with Gradient Descent

For numerical experiments, the state space $\mathcal{X}$ will be discretized and we will denote by $N$ its cardinality. The cost will be represented by a matrix $C \in \mathbb{R}_+^{N \times N}$. The solution to MCOT-C obtained in Proposition 1 may be expressed as

$$\pi_{i,j}^* = u_i G_{i,j} \exp\left( -\zeta^{*\intercal} f_j \right), \tag{10}$$

where $G$ is the Gibbs kernel defined by $G_{i,j} = \exp(-C_{i,j}/\varepsilon)\mu_{2,j}$ and $u_i = \mu_{1,i} / \sum_j G_{i,j} e^{-\zeta^{*\intercal} f}$. As shown in Proposition 2, it is possible to obtain a gradient descent algorithm 1, which looks similar to the Sinkhorn algorithm (Cuturi, 2013), the difference being the update of $\zeta^k$.

---

**Algorithm 1** Semi-Sinkhorn with Gradient Descent

---

   **Input:** $\mu_1$, $C$, $f$
   $\zeta^0 \leftarrow \mathbf{0_M}$
   $k \leftarrow 0$
   **while** $k \leq k_{max}$ **do**
     $u_i^{k+1} \leftarrow \mu_{1,i}/\sum_j G_{i,j}e^{-\zeta^{k\intercal}f}$
     $\zeta^{k+1} \leftarrow \zeta^k + \sum_{i,j} f_j u_i^{k+1} G_{i,j}e^{-\zeta^{k\intercal}f}$
     $\zeta^{k+1} \leftarrow \max\{0, \zeta^{k+1}\}$
     $k \leftarrow k+1$
   **end while**

---

It is also possible to perform Newton's method rather than gradient descent by changing the update of $\zeta^k$ by

$$\zeta^{k+1} \leftarrow \zeta^k + (\Sigma^{\varepsilon\zeta^k})^{-1} \sum_{i,j} f_j u_i^k G_{i,j}e^{-\zeta^{k\intercal}f},$$

where $\Sigma^{\varepsilon\zeta^k}$ is the Hessian defined in equation 9b. In cases where the starting point $\zeta^0$ is close to the optimum $\zeta^*$, we can obtain quadratic convergence (Kelley, 1999).

## 3 Use case: EV charging

### 3.1 Presentation of the use case

Consider a large fleet of electric vehicles (EVs) arriving to a charging station at random times and with random state of charge, according to an initial law $\nu_0$. There is a central planner whose goal is to maintain constraints for the aggregate power consumption, as well as constraints for each vehicle owner. The vehicles arrive during the period $[9\text{am}, 10:30\text{am}]$, and must be fully charged by 5pm.

The goal is power tracking: total power consumption should follow a reference signal $(r_t)$ over a time period $[t_1, t_2]$, with $9\text{am} \leq t_1 < t_2 \leq 5\text{pm}$. This objective arises from the need to ensure real-time balance between power production and demand in electricity grids, where maintaining frequency stability requires aggregate consumption to closely follow regulation signals (Srivastava et al., 2022). This can be formulated as an MCOT-C problem over the space of distributions on $\mathcal{X} = \mathcal{S} \times \mathcal{W}$ with $\mathcal{S} = [0,T] \times [0,1]$ and $\mathcal{W} = [0,T]$. The two first coordinates of $x \in \mathcal{X}$ are the time and the battery state of charge at the arrival and the third is the time when the EV will start charging, called the *plugging time*; so $x \in \mathcal{X}$ is of the form $x = (t_a, b, t_c)$. The function $f$ is defined as:

$$\forall x = (t_a, b, t_c) \in \mathcal{X}, t \in [0,T], f_t(x) = \begin{cases} p \text{ if } t \in [t_c, t_c + \frac{1-b}{v}] \\ 0 \text{ otherwise} \end{cases},$$

where $p$ is the power consumption (here we normalize it to 1) and $v = 0.25h^{-1}$ is the speed of charge of the EVs.

At each iteration, a gradient is calculated on $\mathcal{X} \times \mathcal{W}$, with complexity of computing the gradient at each iteration of the algorithm $O(n_{\text{time}}^3 \times n_{\text{battery}})$, with $n_{\text{time}} = 25$ and $n_{\text{battery}} = 20$, being the number of

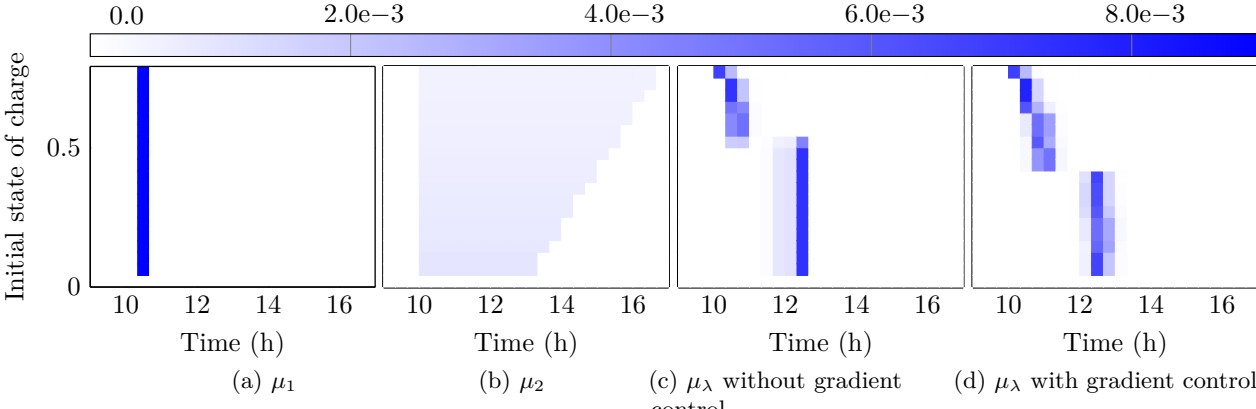

Figure 1: For vehicles arriving at 10am : (a) $\mu_1$ (b) $\mu_2$ designed to encode physical and quality of service constraints; (c) optimized $\mu$ without gradient control; (d) optimized $\mu$ with gradient control.

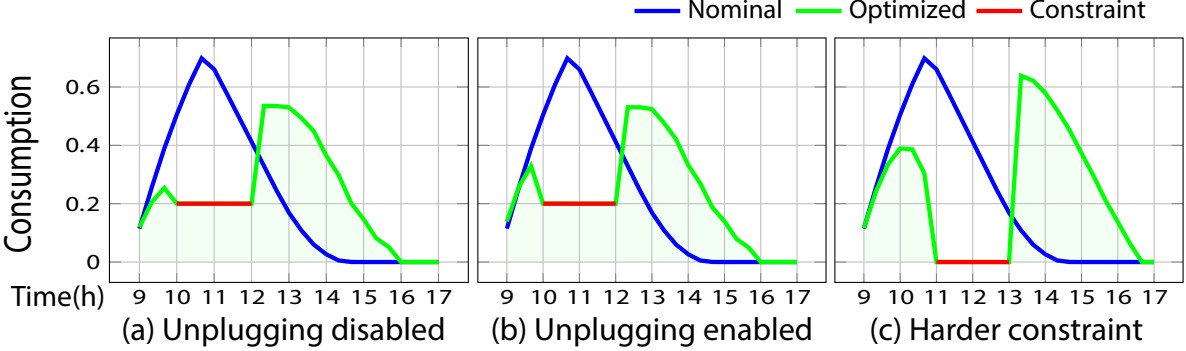

Figure 2: (a) optimized consumption compared to the nominal with unplugging disabled; (b) optimized consumption with unplugging enabled; (c) optimal consumption with constraint infeasible without unplugging.

discretization points in time and battery state of charge. We use the MCOT-C problem presented in Section 2 with $\varepsilon = 0.03$ being a compromise between computational stability and having a low value (as any non-negative value will enforce the physical constraints). This regularizer penalizes the entropic discrepancy between $\pi$ and $\mu_1 \otimes \mu_2$. As a result, it pushes the optimal policy $\pi_2$ to remain close to $\mu_2$, which will later be chosen as a uniform distribution. We consider a version of problem MCOT-C with $\mu_1$ modeling the naive decision rule in which a vehicle initiates charging on arrival:

$$\mu_1(t_a, b, t_c) = \left\{ \begin{array}{l} \nu(t_a, b) \text{ if } t_a = t_c \\ 0 \text{ otherwise} \end{array} \right. .$$

Initiation of charging must be after the arrival time (physical constraint) and every vehicle must be fully charged no later than 5pm (quality of service constraint). The following distribution meets these requirements, $\mu_2(t_a, b, t_c) = \mathbf{Unif}_{[t_a, T - \frac{1-b}{v}]}(t_c)$, with $v$ being the charging speed and $\mathbf{Unif}_{[a,b]}$ being the density of uniform distribution over $[a, b]$. It is assumed that drivers wish to initiate charging as soon as possible: this makes it easier for the driver to manage an unforeseen event and may make it easier for the central planner to respond to a grid contingency. This preference is modeled through the cost $c((.,., t_c^x), (.,., t_c^y)) = (t_c^x - t_c^y)^2$.

### 3.2 Numerical Results

In this section, we present the numerical results, focusing on the satisfaction of the moment constraints as well as on their impact on the optimized distributions. The results for each use case are provided and discussed in Appendix E. In addition, a numerical comparison with a method based on the resolution of the Hamilton–Jacobi–Bellman equations is provided in Appendix F, to illustrate the computational complexity of Algorithm 1.

**EV charging without unplugging** The first results described here impose an additional constraint: once charging begins, it cannot be interrupted until the vehicle is fully charged. In the following simulations, a constraint on power consumption is imposed for the time period beginning at $t_1 = 10$am and ending at $t_2 = 12$pm. As the optimizer $\mu^*$ will be mutually absolutely continuous with respect to $\mu_2$, both physical constraints and constraints on quality of service are imposed through choice of $\mu_2$. In Figure 1(b), the constraints enforced on $\mu_2$ can be observed:

- Quality of Service constraint: At 5 pm, all EVs must be fully charged. Thus, if a vehicle needs $\Delta t$ minutes to charge, then the probability of connecting between 5pm$-\Delta t$ and 5pm is zero. This is observed by the completely white lower right triangle.

- Physical constraint: Vehicles cannot charge before arriving, so there is no mass probability before 10am for vehicles arriving at 10am.

These constraints are found in the $\mu_\lambda$ showed in Figure 1(c) and 1(d), as $\mu_\lambda$ is a reweighting of $\mu_2$. Aggregated consumption displayed in Fig. 2 (a) shows that the first vehicles to arrive will start charging, but most of

those arriving just before 10:00 am will initiate charging only if they arrive with a high battery level so that they are fully charged before the start of the constraint window from 10:00 am to 12:00 pm.

**Gradient control to flatten the curve** For real-life applications, controlling overall consumption over part of the day through equality of consumption to a predefined signal can lead to a peak when the constraint is released. This phenomenon, due to the penalization of distant charging times, is observed in the different plots of Fig. 2. Consumption can be smoothed by introducing the derivative constraints

$$\forall t \in [0, T], |\langle g_t, \mu \rangle| \le g_{\max},$$

where $g_t = f_{t+1} - f_t$.

In this example, $g_{\max} = 0.2$, thus the overall consumption must not increase by more than 0.2 per hour, which is what we observe in Fig. 3: consumption at 12pm increases more slowly. We can also see the impact of the constraint on the gradient by looking at the difference between Figure 1(b) and 1(c). In both cases, vehicles arriving with a high battery level are put to charge first. This comes from the quadratic penalty on the start of the charging time: We prefer to charge those which will quickly be completely charged and which will free up space for those which will take longer.

**EV charging with unplugging** The model can be extended by authorizing a vehicle to interrupt and restart charging. In this case, $\mathcal{X}$ is extended with two extra time dimensions corresponding to an unplugging time and a re-

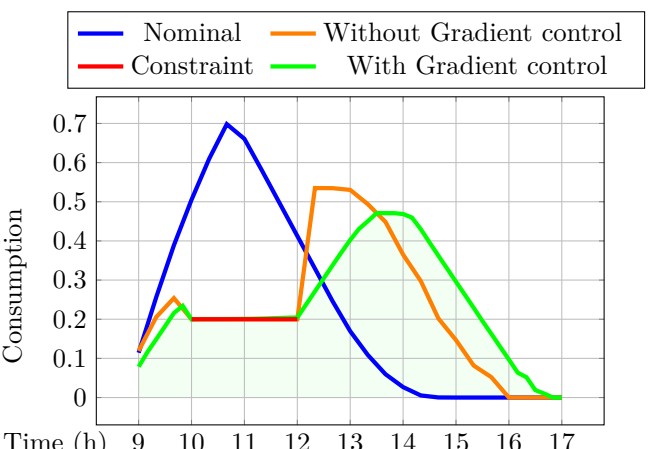

Figure 3: Optimal consumption with and without gradient control of the overall consumption

plugging time. A second term is included in $c$ that is quadratic in the difference of these times, designed to discourage charging interruption.

We find that unplugging does not impact significantly the optimal solution. Fig. 2 (a) and (b) provide a comparison. Only a slight difference is visible before 10 am: A number of vehicles start to charge before the constraint, stop at 10 am and restart afterwards. However, in some cases, this extra flexibility in charging is necessary to obtain a feasible solution. Fig. 2 (c) shows results obtained when power consumption is not permitted in the middle of the day. In any feasible solution, a portion of vehicles stop charging for a period before they are fully charged.

Comparison with piecewise deterministic Markov decision process approach for EV charging control proposed by (Séguret et al., 2024) is provided in Appendix F.

## 4 Online MCOT-C for EV charging

In this section, we provide an online version of MCOT-C and test it on a real dataset.

### 4.1 Formulation of Online MCOT-C

First, while some theoretical models assume perfect knowledge of the battery level at each time step (Séguret, 2023), this value is hard to obtain in practice even if estimates are available (Rezvanizaniani et al., 2014) and existing datasets do not take this data into account (Amara-Ouali et al., 2021). Our choice on this subject is to focus on the leaving time $t_l$ and the charging need $\Delta t_n$, which is the charging time requested by the EV owner. These parameters are easier to access and are consistent with other articles studying real datasets (He et al., 2012; Sadeghianpourhamami et al., 2018). Arriving EVs are therefore defined on the following state space:

$$\mathcal{S} = \underbrace{[0,24]}_{\substack{\text{Arriving time} \\ t_a}} \times \underbrace{[0,24]}_{\substack{\text{Leaving time} \\ t_l}} \times \underbrace{[0,24]}_{\substack{\text{Charging need} \\ \Delta t_n}} \times \underbrace{\{1,\ldots,n_{\text{power}}\}}_{\substack{\text{Max power} \\ p_{max}}}. \tag{11a}$$

At each time step $t \in [0,24]$, EVs are controlled through their charging starting time $t_c$. The control space is thus defined as:

$$\mathcal{W}^{(t)} = \underbrace{[t,24]}_{\text{Plugging time } t_c}, \tag{11b}$$

and we define the product space: $\mathcal{X}^{(t)} = \mathcal{S} \times \mathcal{W}^{(t)}$. At each time step $t \in [0,24]$,

1. New EVs arrive at the charging station and are added to the list of vehicles already present and not charging yet $\{S_i^{(t)}\} = \{S_i : t_a^i \le t \text{ and } t_c^i \ge t\}$. The empirical $\nu^{(t)}$ is updated:

$$\nu^{(t)}(s) = \begin{cases} \frac{1}{N_t} \sum_i \delta(s - S_i^{(t)}) & \text{if } t_a \le t \\ \frac{N}{N_t} \nu(s) & \text{if } t_a > t \end{cases}, \tag{12a}$$

where $N_t = \int_{\mathcal{S}} \sum_i \delta(s - S_i^{(t)}) ds + N \int_{\mathcal{S}} \nu(s) \mathbf{1}_{t_a > t}(s) ds$ is the number of vehicles already arrived and not charging plus the number of vehicles that are estimated to arrive.

2. $\mu_1^{(t)}$ is defined by the "Plug when Arrive" strategy: $\forall s = (t_a, t_l, \Delta t_n, p) \in \mathcal{S}$,

$$\mu_1^{(t)}(s, t_c) = \nu^{(t)}(s)\delta(t_c - t_a). \tag{12b}$$

3. $\mu_2^{(t)}$ is defined as "Plug with a uniform distribution" strategy:

$\forall s = (t_a, t_l, \Delta t_n, p) \in \mathcal{S}, t_c \in \mathcal{W}$,

$$\mu_2^{(t)}(s, t_c) = \begin{cases} \mathbf{Unif}_{[t_a, t_l - \Delta t_n]}(t_c)\nu^{(t)}(s) & \text{if } t_a > t \\ \mathbf{Unif}_{[t, t_l - \Delta t_n]}(t_c)\nu^{(t)}(s) & \text{if } t_a \le t \end{cases}, \tag{12c}$$

where $\mathbf{Unif}[a,b]$ is the density of the uniform distribution on the segment $[a,b]$. For the sake of simplicity, we assume that there is no outlier (no vehicle that would require more charging time than the difference between their arrival time and leaving time in particular). As in Section 3, $\mu_2$ is designed to incorporate the hard constraint of respecting the quality of service through the absolute continuity of $\mu$ with $\mu_2$ (due to the KL term).

4. The central planner will minimize equation 5 to obtain:

$$\pi^{(t)} = \underset{\substack{\pi \in K(\mu_1^{(t)}) \\ \pi_2 \in \mathcal{P}_{f^{(t)}}}}{\arg\min} \langle \pi, c \rangle + \varepsilon D_{\text{KL}}(\pi || \mu_1^{(t)} \otimes \mu_2^{(t)}).$$

The function $c$ chosen here is a quadratic penalization: $c((s^x, t_c^x), (s^y, t_c^y)) = (t_c^x - t_c^y)^2$. In this case, as we compare it with the "Plug when Arrive" strategy for which $t_c^x = t_a^x$, $c$ is a penalty for starting charging long after the vehicle arrives.

5. For each vehicle $S_i^{(t)}$, its plugging time $t_c^i$ is randomly chosen according to $\pi_2^{(t)}(S_i^{(t)}, .)$. $f$ is then updated as: $f^{(t+1)} = f^{(t)} + \frac{1}{N} \sum_{t_c^i = t} f(S_i^{(t)})$. Vehicles $S_i^{(t)}$ such that $t_c^i = t$ begin their charging.

## 4.2 Algorithm

In Algorithm 2, $Alg(\zeta^{(t)}, \mu_1, \mu_2)$ returns $\zeta^{(t+1)}$ the value of Algorithm 1 with the stopping criterion $N_t \|(\langle f^{(t)}, \mu_{\zeta^{(t)}} \rangle)^+\| \leq N\kappa$ and $(.)^+$ is the positive part function: $\forall x \in \mathbb{R}^M, (x)_m^+ = \max(0, x_m)$. The norm $\|\|$ can be chosen as desired, but a good candidate is the infinite norm. With this norm, $N\kappa$ corresponds to the maximum error on all the vehicles that we can afford to have, so it should be chosen relatively small. We can estimate that this error evolves linearly with N, which explains the multiplication by $N$ (as N is the order of magnitude of the vehicles that will arrive during the day). We define the convergence error at time $t$ as $\mathcal{E}_t(\zeta) = \frac{N_t}{N} \|(\langle f^{(t)}, \mu_{\zeta^{(t)}} \rangle)^+\|$ and $\nu_r$, the real arrival law of EVs. With the definitions of $\mu_2^{(t)}$ and $\mu_1^{(t)}$ in equation 12 and Proposition 1, we define $F_\zeta$ as: $\forall s \in \mathcal{S}$, $F_\zeta(s) =$

$$\begin{cases} \dfrac{\int_{\mathcal{W}} \mu_\zeta^{(t)}(s, t_c) f(s, t_c) dt_c}{\nu^{(t)}(s)} & \text{if } \nu^{(t)}(s) \neq 0 \\ 0 & \text{otherwise} \end{cases}.$$

---

**Algorithm 2** Online MCOT-C

**Input:** $\nu$, $N$, $(f_m)_{1 \leq m \leq M}$, $\kappa$
**Output:** V= {} the list of vehicles with their plugging time
S← {}
$\zeta^0 \leftarrow \mathbf{0_M}$
**for** $t$ **from** $0$ **to** $T$ **do**
    Add to S, vehicles that arrived at time $t$
    Compute $N_t$
    Update $\nu$, $\mu_1$ and $\mu_2$ as in equation 12
    $\zeta_m \leftarrow Alg(\zeta, \mu_1, \mu_2, y)$
    **for** $S_i$ in S **do**
        $t_c$ is generated according to $\text{Mu}(\zeta, \mu_1, \mu_2, (S_i, .))$

        **if** $t_c = t$ **then**
            $f \leftarrow f - \frac{1}{N} f(S_i)$
            $S_i$ is removed from S and $(S_i, t_c)$ is added to V
        **end if**
    **end for**
**end for**

---

**Proposition 3. (i)** $\mathcal{E}_{t+1}(\zeta_t)$ *is bounded by* $\kappa$, *a stochastic term, and a term corresponding to a poor prediction of the law* $\nu$:

$$\mathcal{E}_{t+1}(\zeta_t) \leq \kappa + \left\| \left( \sum_{t_a^i = t+1} \frac{F_\zeta(S_i^{(t+1)})}{N} - \mathbb{E}_{\nu_r}[F_\zeta \mathbf{1}_{t_a = t+1}] \right)^+ \right\| + \left\| \left( \mathbb{E}_{\nu_r}[F_\zeta \mathbf{1}_{t_a = t+1}] - \mathbb{E}_\nu[F_\zeta \mathbf{1}_{t_a = t+1}] \right)^+ \right\|.$$

**(ii)** *The second term could be bounded with the Bienaymé–Chebyshev inequality to obtain:*

$$\mathbb{P}\left( \left\| \left( \sum_{t_a^i = t+1} \frac{F_\zeta(S_i^{(t+1)})}{N} - \mathbb{E}_{\nu_r}[F_\zeta \mathbf{1}_{t_a = t+1}] \right)^+ \right\| \geq \kappa_0 \right) \leq \frac{\mathbb{V}_{\nu_r}[F_\zeta \mathbf{1}_{t_a = t+1}]}{N \kappa_0^2}.$$

Thus, there is no need to start the optimization from scratch at each time step, as the solution from the previous step provides a natural warm start for $\zeta$. This starting point is better if (i) the estimation of the arrival law of the vehicles $\nu$ is close from the real arrival law of vehicles $\nu_r$ and (ii) if $N$, the order of magnitude of EVs is large.

## 4.3 Data Overview

The dataset used in this paper is composed of 10.000 charging sessions from public charging stations operated by EVnetNL in the Netherlands (OpenDataset, 2019), in the year 2019. For each charging session, several pieces of information are provided including the arrival time $t_a$, the leaving time $t_l$, the plugging time $\Delta t_n$, and the max power $P$. A more detailed description can be found in (Refa & Hubbers, 2019), and this dataset has already been used for clustering algorithms (Straka & Buzna, 2019), but not yet for MFC.

There is a difference between weekdays and weekend days, so in this paper, we will consider the 7253 charging sessions happening during weekdays and divide them randomly. 90% of these weekdays will form a training set of 231 days (6540 charging sessions) and will be considered historical data. A test day is created with the remaining 10% of weekdays (21 days : 674 charging sessions) by grouping the corresponding 713 vehicle arrivals. The predicted distribution $\nu$ is computed on the training set considered historical data and $N = \frac{6210}{9} = 690$ is the number of vehicles expected to arrive on this test day. In equation 5, we set $\varepsilon = 0.1$ because we want a relatively low value to limit the impact of entropic relaxation (term in Kullback-Leibler),

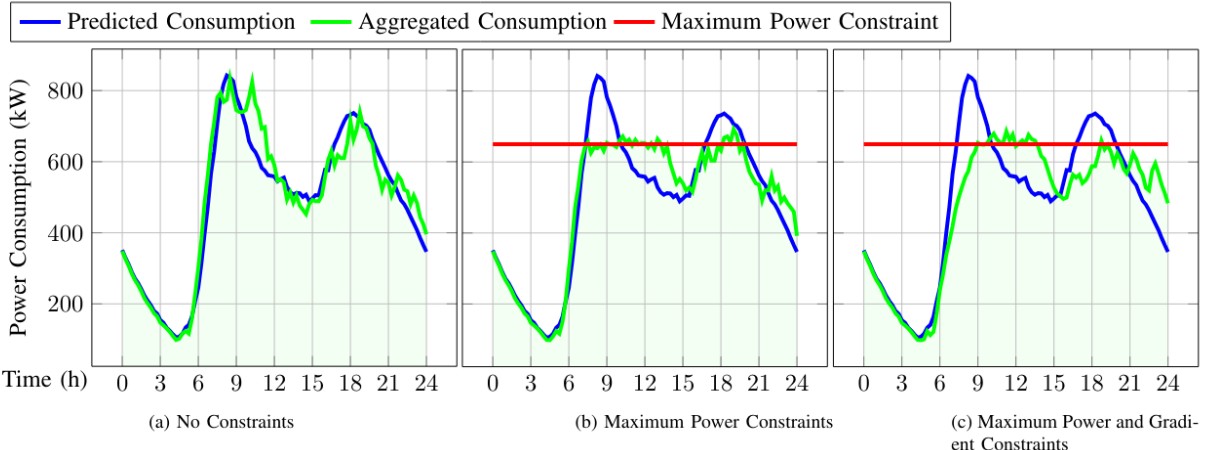

Figure 4: (a) Consumptions for the "Plug when Arrive" $\mu_1$ strategy with the arrival of EV predicted with $\nu$ and with the real distribution of EV; (b) Optimized Consumption for a constraint of 650kW for the aggregated consumption; (c) Optimized consumption for the same maximum power constraint and a constraint of 120kW/h for the gradient of the aggregated consumption.

but not too low, as this risks posing computational problems (because of the $\varepsilon^{-1}$ in the exponential in Proposition 1.

To compute the gradient at each iteration of Algorithm 1, we need to discretize the state space $\mathcal{X}$: The day is divided into $T + 1 = 97$ steps (indexed from 0 to $T$) with a stepsize $\Delta t$ of 15 minutes, which allows rapid grid constraint changes to be taken into account. For the power discretization, we group each EV between 4kW, 7.5kW, and 12kW. This choice of discretization is standard (used for example in (Sadeghianpourhamami et al., 2018)). We assume here that vehicles plugged the day before are not affected by our strategy, because they are already connected, but their consumption is taken into account in order to come closer to reality, particularly in the case of controlling the gradient of aggregate consumption. We therefore consider the aggregate consumption of vehicles arriving throughout the day and that of vehicles arriving the day before (this impact is mainly present before 8 a.m.).

### 4.4 Control of the aggregated consumption

On Fig. 4, the nominal consumption in blue corresponds to what is expected by the charging station, these are the historical data with the plugging strategy $\mu_1$ "Plug when Arrive". On (a), we can see the difference with the consumption for the real arrival of EV during the day with the same plugging strategy. The first peak in the morning lasts longer, while the second peak seems to be weaker. On (b), a constraint imposed by the charging station over the power consumed of $r_f = 650$kW is added through the moment constraints: define for each $m$ the function $f_m$ via $f_m(s, t_c) = p_{\max}$ if $m \in [t_c, t_c + \Delta t_n]$, $f_m(s, t_c) = 0$ otherwise, and impose for each $m$ the constraint $\langle f_m, \mu \rangle - r_f \leq 0$.

This value of 650kW is chosen arbitrarily here, and any other can be chosen as long as it remains realistic. This optimization makes it possible to exploit flexibility while respecting the imposed constraint, despite the prediction error on the length of the first peak. Peaks above the maximum constraint correspond to unforeseen arrivals of a large number of vehicles that must connect directly. It can also be due to the convergence not completely achieved by the algorithm, which depends on the value of $\kappa$ here chosen at 10kW.

### 4.5 Control of the gradient of the aggregated consumption

Another constraint that we want to respect in order to preserve the grid stability is the speed with which consumption will increase or decrease. On Fig. 4 (a) (b), we see a strong peak at the start of the day. We will seek to smooth this peak by imposing a constraint on the gradient of the power consumed. On (c), this constraint imposed by the charging station of $r_g = 100$kW/h is added through the moment constraints:

$\forall m \in [0, T-1], \forall (s, t_c) \in \mathcal{X}^{(t)}, g_m(s, t_c) = f_{m+1}(s, t_c) - f_m(s, t_c)$ and we impose: $\forall m \in [0, T-1], -r_g \leq N \langle g_m, \mu \rangle \leq r_g$.

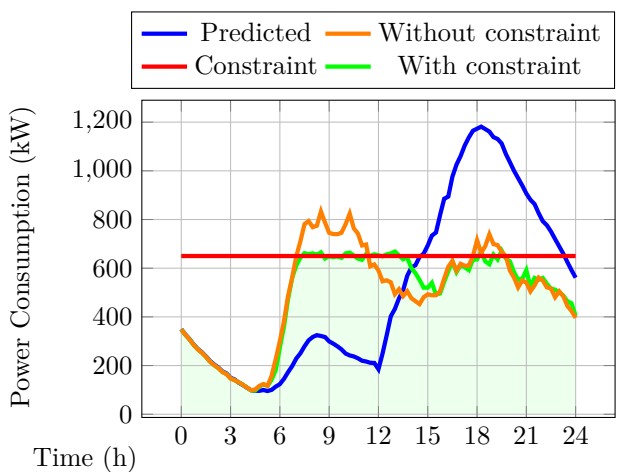

Figure 5: When the prediction $\nu$ differs greatly from the reality

This addition of constraints makes it possible to smooth out the slope which begins around 6am. There are always irregularities due to deviation from prediction and the slight excess of the constraint on the first peak can be explained by the maximum exploitation of the flexibility of the vehicles to respect the gradient constraint, which does not leave enough flexibility when vehicles arrive between 9am and 3pm and have to be connected directly.

### 4.6 Sensitivity to the difference between actual EV arrival and its prediction

This model depends on the quality of the prediction $\nu$ made for the rest of the day. In this part, we try to test the robustness against this quality of prediction, by twisting the previous prediction: the central planner expects 30% less vehicles before 12am and 30% more vehicles after. The aggregated power consumption associated to this prediction is shown in blue in Fig. 5. We can thus observe that compliance with the same maximal power constraint of $650kW$ is still obtained and the consumption is very close to Fig. 4 (b). We therefore have a certain robustness of the model concerning the prediction $\nu$. This robustness is surely obtained here by the fact that we can change the connection time of a previously arrived vehicle as long as it is not connected. The algorithm can therefore, in the event of an unexpected arrival of vehicles to be connected immediately, postpone the connection time of less priority vehicles. But this poorer prediction comes at a cost: when comparing $\langle \pi, c \rangle$ between the case where the prediction is close (shown in figure 4 (a)) and this case, we find that the average time between arrival time $t_a$ and connection time $t_c$ increases from 11 minutes to 12 minutes. Having a less accurate prediction will therefore make less optimal use of flexibility.

### 4.7 Comparison with a non-predictive algorithm

Other algorithms and methods have been proposed in the literature for charging electric vehicles while respecting global constraints, such as IOCS (Integral Online Charging Station) (Alinia et al., 2022). Compared to this algorithm, our approach allows two new things. Firstly, the formulation as a MFC problem allows us to scale up to a very large number of vehicles. Thus, IOCS has a complexity in $O(N^2)$ with $N$ the number of vehicles, whereas ours has a linear complexity $O(N)$. Also, the addition of a prediction allows us to find better solutions. In Fig. 6, we compare our MCOT method with IOCS modified to have the same control (plugging at a given instant). Here, we assume that the

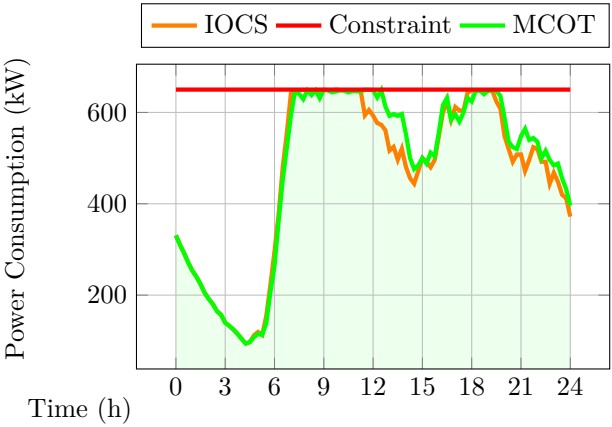

Figure 6: Comparison with IOCS (no prediction)

global constraint cannot be exceeded and that vehicles that cannot be plugged will be rejected. All vehicles have the same priority to connect, and our metric for comparing the two algorithms will therefore be the number of vehicles accepted with the same maximum power constraint of 650kW. On this dataset, MCOT rejects 23 vehicles (3.4% of EVs) while IOCS rejects 33 vehicles (4.9% of EVs). In particular, we see a difference between noon and 3 p.m., when the prediction seems to allow more vehicles to be charged.

## 5   Conclusions

One-sided moment relaxation of OT problem provides a very natural representation setting for MFC applications. This framework considers problems where the initial distribution is known and the goal is to reach a final distribution that satisfies moment constraints, while minimizing a certain control cost. A direct application is found in electric vehicle charging, where the objective is to optimally schedule charging to control their aggregate consumption. It could also be applied to more complex problems in which the distributions represent distributions of trajectories that are solutions of differential equations, for instance, the temperature evolution of a water heater (as presented in Appendix G). In such contexts, MCOT-C ensures that trajectories in the optimal policy $\mu$ are necessarily solutions of this ODE, through absolute continuity. Beyond demand response, other potential applications can be envisioned, such as controlling a population of drones to provide flexible network coverage services (Chen et al., 2020). In such applications, the OT problem is often infinite-dimensional (e.g. trajectories of agents). By introducing an entropic regularization that allows for obtaining an explicit expression of the gradient, MCOT-C leads to a tractable algorithm. Furthermore, KL-term has a dual role in MCOT-C: a relaxation term as in many other machine learning algorithms, but it also enables to enforce the constraints on the dynamics via the choice of $\mu_2$ and absolute continuity imposed by KL. There are many directions for future research:

• The "Semi-Sinkhorn" algorithm might be improved through the introduction of optimization techniques such as proximal methods or momentum.

• In some problems, the size of state space $\mathcal{X}$ is very large (e.g. cases where a continuous $\mathcal{X}$ space cannot be discretized, as in Appendix G). It can also arise when the control becomes more complex (for instance, by allowing unplugging and replugging times in Section 3). As the complexity of the algorithm increases with the size of this state space, it may be necessary to adapt this method to limit computation time, by using Monte Carlo-type methods, i.e., generating a number of trajectories to obtain an approximation of the gradient, instead of calculating it exactly.

• We believe that representing distributions by their moments to perform optimal transport has broader applications in machine learning and control. We aim to explore its potential in other contexts.

### Acknowledgments

This work was carried out in the framework of the AI-NRGY project, funded by France 2030 (Grant No: ANR-22-PETA-0004). Financial support from ARO award W911NF2010055 and NSF award CCF 2306023 are gratefully acknowledged.

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

In this appendix, dualization and proofs are presented in Section A and B. A theoretical extension is presented in appendix C, in the case where the distributions are Gaussian and the moments specified are the means and variances. In appendix D, an experiment involving the transport of a uniform law illustrates the convergence of the regularized problem to the non-regularized problem, when the regularization parameter $\varepsilon$ tends to 0. In Appendix E, the value of the cost function is reported for the different constraints presented in Section 3. In Appendix F, a comparison is done with an Hamilton Jacobi method on the EV use case. Lastly, in Appendix G, an application to the case of water heaters control is proposed.

## A  Duality

First, we want to introduce 2 preliminary problems to the MCOT-C problem. The first problem is a variant of the relaxation of (Alfonsi et al., 2020):

**Problem 1S-MCOT:** *One Sided Moment Constrained Optimal Transport.*

$$d(\mu_1, \mathcal{P}_f) = \min\big\{\langle \pi, c \rangle : \pi \in \mathcal{U}(\mu_1, \mu), \ \mu \in \mathcal{P}_f\big\}. \tag{13}$$

Problem 1S-RMCOT is regularized using Kullback-Leibler divergence:

**Problem 1S-RMCOT:** *One Sided - Regularized Moment Constrained Optimal Transport (1S-RMCOT).*

$$d_\varepsilon(\mu_1, \mathcal{P}_f) = \min_{\mu,\pi}\big\{\langle \pi, c \rangle + \varepsilon D_{\mathrm{KL}}(\pi \| \mu_1 \otimes \mu_2) : \pi \in \mathcal{U}(\mu_1, \mu), \ \mu \in \mathcal{P}_f\big\}, \tag{14}$$

where $\varepsilon > 0$.

### A.1  Dual for 1S-MCOT

Characterization of a solution to Problem 1S-MCOT is based on a Lagrangian relaxation. Introduce two classes of Lagrange multipliers for equation 13: $\psi$ is for the first marginal constraint, a real-valued measurable function on $\mathcal{X}$, and $\lambda \in \mathbb{R}_+^M$ for the moment constraints. The dual functional is defined as the infimum,

$$\varphi^*(\psi, \lambda) := \inf_\pi \ \langle \pi, c \rangle - \langle \pi_1 - \mu_1, \psi \rangle + \langle \pi_2, \lambda^\mathsf{T} f \rangle = \langle \mu_1, \psi \rangle + \inf_{x,y}\{c(x,y) - \psi(x) + \lambda^\mathsf{T} f(y)\}. \tag{15}$$

The convex dual of equation 13 is defined to be the supremum of $\varphi^*(\psi, \lambda)$ over all $\psi$ and $\lambda$. The dual optimization problem admits a familiar representation. Compactness is assumed in Proposition 4 (ii), as in prior work (Kemperman, 1968).

**Proposition 4.** *If (A1) and (A2) hold, then,*

**(i)** *With $\varphi^*$ defined in equation 15, the dual convex program admits the representation*

$$d^* := \sup_{\psi,\lambda} \varphi^*(\psi, \lambda) = \sup_{\psi,\lambda}\big\{\langle \mu_1, \psi \rangle : \psi(x) - \lambda^\mathsf{T} f(y) \leq c(x,y) \ \text{ for all } x, y\big\}. \tag{16}$$

*On replacing $\psi$ with $\psi^\lambda(x) := \inf_y\{c(x,y) + \lambda^\mathsf{T} f(y)\}$ we obtain the equivalent max-min problem*

$$d^* = \sup_\lambda \int \inf_y [c(x,y) + \lambda^\mathsf{T} f(y)]\mu_1(dx). \tag{17}$$

**(ii)** *Suppose in addition the set $\mathcal{X}$ is compact. Then the supremum in equation 16 is achieved, and there is no duality gap: for a vector $\lambda^* \in \mathbb{R}_+^M$,*

$$d(\mu_1, \mathcal{P}_f) = d^* = \int \min_y\{c(x,y) - {\lambda^*}^\mathsf{T} f(y)\}\mu_1(dx).$$

Once we solve equation 16, we obtain $\pi^*$ through complementary slackness:

$$0 = \int_{x,y} \pi^*(x,y)\{\psi^*(x) + {\lambda^*}^\mathsf{T} f(y) - c(x,y)\},$$

which means that $\pi^*$ is supported on the set $\{(x,y) : -{\lambda^*}^\mathsf{T} f(y) + \psi^*(x) = c(x,y)\}$.

## A.2 Regularization

Recall that the functional $D_{\mathrm{KL}}(\pi\|\mu_1\otimes\mu_2)$ is used to define the Sinkhorn distance (Cuturi, 2013), and coincides with mutual information when the marginals of $\pi$ agree with the given probability measures $\mu_1$ and $\mu_2$. In the present paper, the marginal $\mu_2$ is a design parameter.

**1S-RMCOT geometry and duality** A close cousin to 1S-RMCOT uses the Kullback-Leibler divergence as a constraint rather than penalty (Cuturi, 2013). Consider for fixed $\delta > 0$,

$$d_\delta^c(\mu_1, \mathcal{P}_f) = \min\{\langle \pi, c\rangle, \quad \text{s.t. } \pi \in \mathcal{U}(\mu_1, \mu), \mu \in \mathcal{P}_f, D_{\mathrm{KL}}(\pi\|\mu_1\otimes\mu_2) \le \delta\}. \tag{18}$$

The parameter $\varepsilon > 0$ in equation 14 may be regarded as a Lagrange multiplier corresponding to the constraint $D_{\mathrm{KL}}(\pi\|\mu_1\otimes\mu_2) \le \delta$. Under general conditions there is $\delta(\varepsilon)$ such that the optimizers of equation 18 and equation 14 coincide.

In considering the dual of equation 14 we choose a relaxation of the moment constraints only: letting $\lambda \in \mathbb{R}_+^M$ denote the Lagrange multiplier as before,

$$\varphi^*(\lambda) := \inf_\pi \{\langle \pi, c\rangle + \varepsilon D_{\mathrm{KL}}(\pi\|\mu_1\otimes\mu_2) + \langle \pi_2, \lambda^\intercal h\rangle : \pi_1 = \mu_1\}. \tag{19}$$

The convex dual of 1S-RMCOT is by definition the supremum of the concave function $\varphi^*$. The optimizer, when it exists, is denoted $\pi^\lambda$.

With the notation

$$\ell_0^\lambda(x,y) = -\lambda^\intercal f(y) - c(x,y), \quad x, y \in \mathcal{X}, \tag{20}$$

the dual function may be expressed

$$\varphi^*(\lambda) = -\max_\pi \{\langle \pi, \ell_0^\lambda\rangle - \varepsilon D_{\mathrm{KL}}(\pi\|\mu_1\otimes\mu_2) : \pi_1 = \mu_1\}.$$

The dual of equation 18 with $d = d(\varepsilon)$ yields better geometric insight. If the maximum above exists, then the maximizer $\pi^\lambda$ solves

$$\pi^\lambda \in \arg\max\{\langle \pi, \ell_0^\lambda\rangle : D_{\mathrm{KL}}(\pi\|\mu_1\otimes\mu_2) \le \delta, \ \pi_1 = \mu_1\}.$$

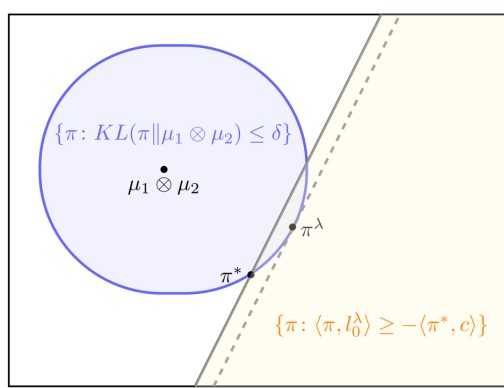

Figure 7: Dual geometry for 1S-RMCOT

The convex region containing $\mu_1 \otimes \mu_2$ shown in Fig. 7 is the set of all $\pi$ for which $\pi_1 = \mu_1$ and $D_{\mathrm{KL}}(\pi\|\mu_1\otimes\mu_2) \le \delta$. The optimizer $\pi^\lambda$ lies on the intersection of this region and the hyperplane shown in the figure, indicated with a dashed line: $\{\pi : \langle \pi, \ell_0^\lambda\rangle = \langle \pi^\lambda, \ell_0^\lambda\rangle\}$. This value of $\lambda$ does not optimize $\varphi^*$ because the hyperplane is not the boundary of the half-space shown in the figure.

For computation, it is convenient to make a change of variables: since $\pi_1 = \mu_1$ is constrained, the infimum is over all probability kernels: for $\lambda \in \mathbb{R}_+^M$,

$$\varphi^*(\lambda) := \inf_T \{-\langle \mu_1 T, \ell_0^\lambda\rangle + \varepsilon D_{\mathrm{KL}}(\mu_1 T\|\mu_1\otimes\mu_2)\} \tag{21}$$

For each $\lambda \in \mathbb{R}_+^M$, $\varepsilon > 0$ and $x \in \mathcal{X}$, we denote

$$B_{\lambda,\varepsilon}(x) = \varepsilon \log \int_{y\in\mathcal{X}} \exp\big(\varepsilon^{-1}\ell_0^\lambda(x,y)\big)\mu_2(dy). \tag{22}$$

**Proposition 5.** *Subject to (A1)–(A3),*

**(i)** *The infimum equation 21 gives $\varphi^*(\lambda) = -\langle \mu_1, B_{\lambda,\varepsilon}\rangle$.*

**(ii)** *The probability kernel maximizing equation 21 is*

$$T^\lambda(x, dy) = \mu_2(dy) \exp(L^\lambda(x, y)), \text{ with } L^\lambda(x, y) = \varepsilon^{-1}\{\ell_0^\lambda(x, y) - B_{\lambda,\varepsilon}(x)\}. \tag{23a}$$

**(iii)** *unique $\lambda^* \in \mathbb{R}_+^M$ exists, satisfying*

$$\varphi^*(\lambda^*) = d_\varepsilon(\mu_1, \mathcal{P}_f). \tag{23b}$$

*That is, there is no duality gap.*

The similarity between Proposition 5 and Proposition 4 is found through examination of equation 17, and the recognition that $-B_{\lambda,\varepsilon}(x)$ is a ($\mu_2$-weighted) soft minimum of $-\ell_0^\lambda(x, y) = c(x, y) - \lambda^\intercal f(y)$ over $y \in \mathcal{X}$. Subject to this interpretation, the convex dual of 1S-RMCOT can be expressed in a form entirely analogous to equation 17:

$$\max_\lambda \varphi^*(\lambda) = \max_\lambda \int \operatorname*{softmin}_y \{c(x, y) + \lambda^\intercal f(y)\}\mu_1(dx).$$

**1S-MCOT approximation**

Consider the following procedure to obtain a solution to 1S-MCOT (without regularization), but with $\mathcal{X}$ compact, and the supports of $\mu_1$ and $\mu_2$ each equal to all of $\mathcal{X}$. Let $\{\pi^\varepsilon, \lambda^\varepsilon : \varepsilon > 0\}$ denote primal-dual solutions to 1S-RMCOT, where $\varepsilon > 0$ is the scaling in equation 14. Hence for each $\varepsilon > 0$,

$$d_\varepsilon(\mu_1, \mathcal{P}_f) = \langle \pi^\varepsilon, c\rangle + \varepsilon D_{\mathrm{KL}}(\pi^\varepsilon \| \mu_1 \otimes \mu_2) = -\langle \mu_1, B_{\lambda^\varepsilon, \varepsilon}\rangle.$$

**Proposition 6.** *Suppose that the assumptions of Proposition 4 (ii) hold, so in particular $\mathcal{X}$ is compact. Then, any weak subsequential limit of $\{\pi^\varepsilon, \lambda^\varepsilon : \varepsilon > 0\}$ as $\varepsilon \downarrow 0$ defines a pair $(\pi^0, \lambda^0)$ for which $\pi^0$ solves 1S-MCOT and $\lambda^0$ achieves the supremum in equation 17.*

*Furthermore, it is possible to bound the rate of convergence:*

$$|d_\varepsilon^*(\mu_1, \mathcal{P}_f) - d^*(\mu_1, \mathcal{P}_f)| \leq \varepsilon D_{KL}(\pi^0 \| \mu_1 \otimes \mu_2)$$

.

## A.3  Link with the MCOT-C Problem

Writing the dual of MCOT-C, we get:

$$\varphi^*(\lambda) = \varepsilon \min_\pi \big\{-\langle \pi, l\rangle + D_{\mathrm{KL}}(\pi \| \mu_1 \otimes \mu_2) : \pi \in K(\mu_1)\big\}.$$

Since $\pi \in K(\mu_1)$ is constrained, the infimum is over all probability kernels $T$ from $\mathcal{X}$ to $\mathcal{W}$:

$$\varphi^*(\lambda) = -\int_x \mu_1(dx) \max_{T(x,.)} \big\{\langle T(x,.), \ell_0^\lambda(x,.)\rangle_\mathcal{W} - \varepsilon D_{\mathrm{KL}}(T(x,.) \| \mu_2(s^x,.))\big\},$$

where $\langle .,.\rangle_\mathcal{W}$ is the inner product on $\mathcal{W}$. We obtain Proposition 1, which gives similar results as Prop. 5 with a probability kernel going from $\mathcal{X}$ to $\mathcal{W}$.

# B  Proofs

Much of the analysis that follows is based on convex duality between relative entropy and log moment generating functions. For any probability measure $\mu$ on $\mathcal{X}$ and function $g: \mathcal{X} \to \mathbb{R}$, the log moment generating function is denoted,

$$\Lambda_\mu(g) = \log\langle \mu, e^g\rangle.$$

With $\mu$ fixed, this is viewed as an extended-valued, convex functional on the space of Borel measurable functions. Lemma 2 is a standard tool in information theory (Dembo & Zeitouni, 1998), and a reason that relative entropy is popular for use as a regularizer in optimization.

**Lemma 2.** *Relative entropy and the log moment generating function are related via convex duality:*

*For any probability measure p we have*

$$D_{KL}(p\|\mu) = \sup_g \{\langle p, g \rangle - \Lambda_\mu(g)\}. \tag{24a}$$

*If $D_{KL}(p\|\mu) < \infty$ then the supremum is achieved, with optimizer equal to the log likelihood ratio, $g^* = \log(dp/d\mu)$.*

*For Borel measurable $g\colon \mathcal{X} \to \mathbb{R}$,*

$$\Lambda_\mu(g) = \sup_p \{\langle p, g \rangle - D_{KL}(p\|\mu)\}. \tag{24b}$$

*If $\Lambda_\mu(g) < \infty$ then the supremum is achieved, where the optimizer $p^*$ has log likelihood ratio $\log(dp^*/d\mu) = g - \Lambda_\mu(g)$.* □

We present here the proof of part (i). The proof of (ii) is done in the proof of Proposition 6.

**Proof of Proposition 4** The dual function is invariant under a constant shift in $\psi$, so we may assume that the infimum is exactly zero by adding a constant to $\psi$. This gives

$$\max_{\psi, \lambda} \phi^*(\psi, \lambda) = \max_{\psi, \lambda} \left\{ \langle \mu_1, \psi \rangle \ : \ \inf_{x,y} [c(x,y) - \psi(x) - \lambda \mid f(y)] = 0 \right\}.$$

The value of the maximum is unchanged if the equality constraint is replaced by the inequality

$$\inf_{x,y} \{ c(x,y) - \psi(x) - \lambda \mid f(y) \} \geq 0,$$

which yields the representation equation 16. □

**Proof of Proposition 5** For each $\lambda$ we have by definition,

$$\varphi^*(\lambda) = \min_T \int_{x \in \mathcal{X}} \mu_1(dx) \left\{ \varepsilon D_{\mathrm{KL}}(T(x, \cdot)\|\mu_2) - \int_{y \in \mathcal{X}} T(x, dy)\ell_0^\lambda(x, y) \right\} \tag{25}$$

$$= -\varepsilon \max_T \int_{x \in \mathcal{X}} \mu_1(dx) \left\{ \varepsilon^{-1} \int_{y \in \mathcal{X}} T(x, dy)\ell_0^\lambda(x, y) - D_{\mathrm{KL}}(T(x, \cdot)\|\mu_2) \right\}. \tag{26}$$

For each $x$ we have an optimization problem of the form equation 24b. Applying Lemma 2 (ii) gives the representation equation 8a and by substitution (or applying equation 24b) we obtain

$$\varepsilon^{-1} \int_{y \in \mathcal{X}} T^\lambda(x, dy)\ell_0^\lambda(x, y) - D_{\mathrm{KL}}(T^\lambda(x, \cdot)\|\mu_2) = \varepsilon^{-1} B_{\lambda,\varepsilon}(x). \tag{27}$$

Integrating with respect to $\mu_1$ and applying equation 26 completes the proof. □

**Proof of Proposition 1** The proof is the same as the previous one using this expression of the dual:

$$\varphi^*(\lambda) = -\int_x \mu_1(dx) \max_{T(x,.)} \left\{ \langle T(x,.), \ell_0^\lambda(x,.) \rangle_{\mathcal{W}} - \varepsilon D_{\mathrm{KL}}(T(x,.)\|\mu_2(s^x,.)) \right\}.$$

**Proof of Proposition 6**   Let $(\pi^\varepsilon, \lambda^\varepsilon)$ denote the solution to 1s-RMCOT, with $\varepsilon > 0$ regarded as a variable. We let $(\pi^0, \lambda^0)$ denote any weak sub-sequential limit: for a sequence $\{\varepsilon_i \downarrow 0\}$,

$$\pi^{\varepsilon_i} \to \pi^0\,, \qquad \lambda^{\varepsilon_i} \to \lambda^0\,, \qquad i \to \infty.$$

Optimality of $\pi^0$ is established in the following steps:

- Subject to (A1) and (A2) we know that $\pi^0 \in \mathcal{U}(\mu_1, \mu)$ with $\mu \in \mathcal{P}_f$.

- For any $\pi \in \mathcal{U}(\mu_1, \mu)$ with $\mu \in \mathcal{P}_f$ and $D_{\mathrm{KL}}(\pi\|\mu_1 \otimes \mu_2) < \infty$ and any $\varepsilon > 0$ we have

$$\langle \pi^0, c\rangle = \lim_{i\to\infty} \langle \pi^{\varepsilon_i}, c\rangle \leq \lim_{i\to\infty}\{\langle \pi^{\varepsilon_i}, c\rangle + \varepsilon_i D_{\mathrm{KL}}(\pi^{\varepsilon_i}\|\mu_1 \otimes \mu_2)\} \leq \lim_{i\to\infty}\{\langle \pi, c\rangle + \varepsilon_i D_{\mathrm{KL}}(\pi\|\mu_1 \otimes \mu_2)\} = \langle \pi, c\rangle\,.$$

- Under the support assumption we can approximate in the weak topology any $\pi \in \mathcal{U}(\mu_1, \mu)$ with $\mu \in \mathcal{P}_f$ by $\pi^\delta$ satisfying $D_{\mathrm{KL}}(\pi^\delta\|\mu_1 \otimes \mu_2) < \infty$ and

$$\langle \pi^0, c\rangle \leq \langle \pi^\delta, c\rangle \leq \langle \pi, c\rangle - \delta\,.$$

Since $\delta > 0$ is arbitrary this establishes optimality.

We next show $\lambda^0$ provides an optimal solution. Then, for any $\lambda$,

$$\langle \pi^0, c\rangle \geq -\lim_{i\to\infty} \langle \mu_1, B_{\lambda,\varepsilon_i}\rangle = \int \in f_y\{c(x,y) - \lambda^T f(y)\}\mu_1(dx)\,.$$

The lower bound is achieved using $\lambda^0$ by allowing $\lambda$ to depend on $i$:

$$\langle \pi^0, c\rangle \leq \lim_{i\to\infty}\{\langle \pi^{\varepsilon_i}, c\rangle + \varepsilon_i D_{\mathrm{KL}}(\pi^{\varepsilon_i}\|\mu_1 \otimes \mu_2)\} = -\lim_{i\to\infty} \langle \mu_1, B_{\lambda^{\varepsilon_i},\varepsilon_i}\rangle = \int \inf_y\{c(x,y) - \lambda^{0^T} f(y)\}\mu_1(dx)\,.$$

To prove the rate of convergence, we adapt results from (Luise et al., 2018) in our context. First, we denote $\pi_\varepsilon = argmin[\langle \pi, c\rangle + \varepsilon D_{\mathrm{KL}}(\pi\|\mu_1 \otimes \mu_2)]$ and by optimality of $\pi_\varepsilon$, we obtain: $\langle \pi_\varepsilon, c\rangle + \varepsilon D_{\mathrm{KL}}(\pi_\varepsilon\|\mu_1 \otimes \mu_2) \leq \langle \pi_0, c\rangle + \varepsilon D_{\mathrm{KL}}(\pi_0\|\mu_1 \otimes \mu_2)$

By optimality of $\pi_0$ and positivity of the Kullback-Leibler divergence, we obtain: $\langle \pi_0, c\rangle \leq \langle \pi_\varepsilon, c\rangle \leq \langle \pi_\varepsilon, c\rangle + \varepsilon D_{\mathrm{KL}}(\pi_\varepsilon\|\mu_1 \otimes \mu_2)$

Combining these inequalities, we get:

$$0 \leq \langle \pi_\varepsilon, c\rangle + \varepsilon D_{\mathrm{KL}}(\pi_\varepsilon\|\mu_1 \otimes \mu_2) - \langle \pi_0, c\rangle \leq \varepsilon D_{\mathrm{KL}}(\pi_0\|\mu_1 \otimes \mu_2),$$

$$0 \leq d_\varepsilon^*(\mu_1, \mathcal{P}_f) - d^*(\mu_1, \mathcal{P}_f) \leq \varepsilon D_{\mathrm{KL}}(\pi_0\|\mu_1 \otimes \mu_2).$$

which proves our result.

**Proof of Lemma 1**   Suppose that $v \in \mathbb{R}^M$ is in the null space: $\Sigma^\lambda v = 0$. From the definition equation 9b it follows that

$$0 = v^\intercal \Sigma^\lambda v = \mathbb{E}^\lambda\big[\big\{v^\intercal\big(f(Y) - \mathbb{E}^\lambda[f(Y) \mid X]\big)\big\}^2\big].$$

Equivalently, there is a function $g\colon \mathcal{X} \to \mathbb{R}$ such that

$$v^\intercal f(Y) = g(X) \qquad a.s.\ [\pi^\lambda].$$

The probability measures $\pi^\lambda$ and $\pi^0 := \mu_1 \otimes \mu_2$ are mutually absolutely continuous, so the same equation holds under a.s. $[\pi^0]$. Independence gives

$$v^\intercal f(Y) = \mathbb{E}^0[v^\intercal f(Y) \mid Y] = \mathbb{E}^0[g(X) \mid Y] = \langle \mu_1, g\rangle \qquad a.s.\ [\pi^0].$$

That is, the variance of $v^\intercal f(Y)$ is equal to zero. Under (A3) this is possible only if $v = 0$.   □

**Proof of Proposition 2**   Recall the notation $\mu^\lambda = \mu_1 T^\lambda$, which is the second marginal of $\pi^\lambda$, and the probabilistic notation defined in the Introduction. Also, by definition we have $\mathcal{J}(\zeta) = \varepsilon^{-1}\langle \mu_1, B_{\varepsilon\zeta,\varepsilon}\rangle$.

We have for each $i$,

$$\varepsilon^{-1}\frac{\partial}{\partial\zeta_i}B_{\varepsilon\zeta,\varepsilon}(x) = \frac{\int_{y\in\mathcal{X}}\mu_2(y)\exp\big(\{\zeta^\mathsf{T}f(y) - \varepsilon^{-1}c(x,y)\}\big)f_m(y)}{\int_{y\in\mathcal{X}}\mu_2(y)\exp\big(\{\zeta^\mathsf{T}f(y) - \varepsilon^{-1}c(x,y)\}\big)} = T^\lambda f_m(x).$$

Integrating each side over $\mu_1$ gives equation 9a (recall that $\mu^\lambda = \mu_1 T^\lambda$).

To obtain the second derivative of $\mathcal{J}(\zeta)$ requires the first derivative of the log-likelihood:

$$L_j^{\varepsilon\zeta}(x,y) := \frac{\partial}{\partial\zeta_j}L^{\varepsilon\zeta}(x,y) = \frac{\partial}{\partial\zeta_j}\big[\zeta^\mathsf{T}f(y) - \varepsilon^{-1}B_{\varepsilon\zeta,\varepsilon}(x)\big] = h_j(y) - T^\lambda h_j(x).$$

From this we obtain,

$$
\begin{aligned}
\frac{\partial^2}{\partial\zeta_i\partial\zeta_j}B_{\varepsilon\zeta,\varepsilon}(x) &= \frac{\partial}{\partial\zeta_j}T^{\varepsilon\zeta}f_m(x) \\
&= \int T^{\varepsilon\zeta}(x,dy)\{L_j^{\varepsilon\zeta}(x,y)f_m(y)\} \\
&= \int T^{\varepsilon\zeta}(x,dy)h_j(y)f_m(y) - T^\lambda h_j(x)\int T^{\varepsilon\zeta}(x,dy)h_j(y) \\
&= \mathbb{E}^\lambda[h_j(Y)f_m(Y)\mid X = x] - \mathbb{E}^\lambda[f_m(Y)\mid X = x]\mathbb{E}^\lambda[h_j(Y)\mid X = x].
\end{aligned}
$$

Integrating each side over $\mu_1$ gives equation 9b.   $\square$

**Proposition 7.** *The conditional distribution defined in equation 8a is Markovian: for a collection of probability kernels $\{\check{P}_i^\lambda\}$ parameterized by $x$,*

$$T^\lambda(x,dy) = \nu_0(dy_0)\prod_{i=1}^M \check{P}_i^\lambda(y_{i-1}, dy_i; x). \tag{28}$$

**Proof of Proposition 7**   The proof reduces to justifying equation 28, which is one component of Proposition 8 that follows.

Write $L_i^\lambda(x_i, y_i) = \varepsilon^{-1}\{\lambda_i(\mathcal{U}(y_i) - r_i) - \frac{1}{2}\|x_i - y_i\|^2\}$, and for each $i$ consider the positive kernel,

$$\widehat{P}_i^\lambda(y_{i-1}, dy_i) = P_i(y_{i-1}, dy_i)\exp\big(L_i^\lambda(x_i, y_i)\big).$$

**Proposition 8.** *The conditional distribution defined in equation 8a can be expressed*

$$T^\lambda(x,dy) = \nu_0(dy_0)\exp\big(-\varepsilon^{-1}B_{\lambda,\varepsilon}(x)\big)\prod_{i=1}^M \widehat{P}_i^\lambda(y_{i-1}, dy_i). \tag{29}$$

*Consequently, conditioned on $X = x$, the process $Y$ is of the form equation 28, in which each kernel in the product takes the form,*

$$\check{P}_i^\lambda(y_{i-1}, dy_i; x) = \frac{1}{g_{i-1}(y_{i-1}; x)}\widehat{P}_i^\lambda(y_{i-1}, dy_i)g_i(y_i; x).$$

*The functions $\{g_i : 0 \le i \le M\}$ are defined inductively: $g_M(y_M; x) \equiv 1$, and for $1 \le i \le M$,*

$$g_{i-1}(y; x) := \int \widehat{P}_i^\lambda(y, dy_i)g_i(y_i; x), \quad y \in \mathsf{X}.$$

*This results in $g_0(y_0, x) = \exp\big(\varepsilon^{-1}B_{\lambda,\varepsilon}(x)\big)$.*

**Proof** The representation equation 29 follows from the definition equation 8a and the structure imposed on $h$ and $\mu_1$. It is then immediate that equation 29 can be transformed to equation 28: by construction,

$$\prod_{i=1}^{M} \check{P}_i^\lambda(y_{i-1}, dy_i; x) = \frac{1}{g_0(y_0; x)} \prod_{i=1}^{M} \widehat{P}_i^\lambda(y_{i-1}, dy_i).$$

Since $y_0 = x_0$ by construction, it also follows that

$$\exp\big(\varepsilon^{-1} B_{\lambda,\varepsilon}(x)\big) = g_0(x_0; x).$$

□

## C    Example: Quadratic Constraints & Gaussian Regularizer

Consider the special case in which the function $f$ is designed to specify all first and second moments for $Y$. To solve Problem 2 we adopt the following notational conventions for the Lagrange multiplier: $\mathbb{E}[Y_i] = m_i^1 \longleftrightarrow \lambda_i^1$ and $\mathbb{E}[Y_i Y_j] = m_{ij}^2 \longleftrightarrow \lambda_{i,j}^2$. Of course we have $m_{ij}^2 = m_{ji}^2$ for each $i, j$. The total number of constraints is thus $M = n + n(n+1)/2$. For purposes of calculation it is useful to introduce the symmetric matrices $M_Y^2$ and $\Lambda^2$ with respective entries $\{m_{ij}^2\}$ and $\{\lambda_{ij}^2\}$; similar notation is used for $m_Y$ and $\lambda^1$, the $n$-dimensional vectors with entries $\{m_i^1\}$ and $\{\lambda_i^1\}$.

**Remark:**  In this subsection, the same assumptions as in the rest of the article are not made; in particular, compactness is not assumed, as this property does not hold in the present setting.

Equation (7) gives $\ell_0^\lambda(x,y) = \lambda^\intercal f(y) - c(x,y)$ with

$$\lambda^\intercal f(y) = y^\intercal \Lambda^2 y - \langle \Lambda^2, M_Y^2 \rangle + y^\intercal \lambda^1 - m_Y^\intercal \lambda^1. \tag{30}$$

An explicit solution to problem 1S-RMCOT is obtained when $c$ is quadratic and $\mu_2$ is Gaussian:

**Proposition 9.** *Consider the 1S-RMCOT optimization problem equation 14 in the following special case:* $c(x,y) = \frac{1}{2}\|x - y\|^2$, *and* $\mu_2 = N(0, I)$ *in the regularizer equation 4. Assume that the target covariance* $\Sigma_Y := M_Y^2 - m_Y m_Y^\intercal$ *is positive definite.*

*Then, for each* $\lambda$ *with* $\Lambda^2 < \frac{1}{2}(1 + \varepsilon)I$, *the probability kernel* $T^\lambda$ *is Gaussian: conditioned on* $X = x$, *the distribution of* $Y$ *is Gaussian* $N(m_{T^\lambda}^x, \Sigma_{T^\lambda})$ *with*

$$m_{T^\lambda}^x = \varepsilon^{-1} \Sigma_{T^\lambda}[x + \lambda^1], \quad \Sigma_{T^\lambda} = \big[I + \varepsilon^{-1}[I - 2\Lambda^2]\big]^{-1}. \tag{31}$$

**Proof of Proposition 9**  From equation 30 and using $c(x,y) = \frac{1}{2}\|x - y\|^2$ we obtain an expression for the likelihood $L^\lambda$ appearing in equation 8a:

$$L^\lambda(x,y) = \varepsilon^{-1}\big\{y^\intercal \Lambda^2 y + y^\intercal \lambda^1 - \kappa^\lambda - B_{\lambda,\varepsilon}(x)\big\} - \tfrac{1}{2}(\|x\|^2 - 2x^\intercal y + \|y\|^2)\big\}, \tag{32}$$

with $\kappa^\lambda = \langle \Lambda^2, M_Y^2 \rangle + m_Y^\intercal \lambda^1$. The expression for $T^\lambda$ in equation 8a using $\mu_2 = N(0, I)$ then implies that for any $x$, $T^\lambda(x, dy)$ admits the Gaussian density

$$\tau^\lambda(y \mid x) = \frac{1}{n^\lambda(x)} \exp\big(-\tfrac{1}{2}\|y\|^2\big) \exp\big(\varepsilon^{-1}\{-\tfrac{1}{2}y^\intercal[I - 2\Lambda^2]y + y^\intercal[x + \lambda^1]\}\big), \tag{33}$$

where $n^\lambda(x) = (2\pi)^{n/2} \exp\big(\varepsilon^{-1}\{\kappa^\lambda + B_{\lambda,\varepsilon}(x) + \tfrac{1}{2}\|x\|^2\}\big)$ may be regarded as a normalizing constant.    □

**Remark:** If $\mu_1$ is non gaussian, it is necessary to compute the normalizing constant in the definition of $T^\lambda$:

$$n^\lambda(x) = n^\lambda(x) \int \tau^\lambda(y \mid x)\, dy = \int \exp\left(-\tfrac{1}{2} y^\intercal \Sigma_{T^\lambda}^{-1} y + \varepsilon^{-1} y^\intercal [x + \lambda^1]\right) dy \qquad (34)$$

$$= \sqrt{(2\pi)^d \det(\Sigma_{T^\lambda})} \exp\left(\tfrac{1}{2} \varepsilon^{-2} [x + \lambda^1]^\intercal \Sigma_{T^\lambda}[x + \lambda^1]\right). \qquad (35)$$

Monte-Carlo methods can be used to estimate $\lambda^*$. Denote for each $x$,

$$q^\lambda(x) = \int T^\lambda(x, dy) f(y)\,, \quad m^\lambda(x) = \int T^\lambda(x, dy) f(y) f(y)^\intercal.$$

Each have polynomial entries: $q_i^\lambda$ is a quadratic function of $x$ and $m_{i,j}^\lambda(x)$ is a fourth order polynomial in $x$ for each $i, j$. Thus, one might take

$$\widetilde{m}^{n+1} = q^{\lambda_n}(X_{n+1})\,, \qquad \widetilde{\Sigma}^{n+1} = m^{\lambda_n}(X_{n+1}) - \widetilde{m}^{n+1}[\widetilde{m}^{n+1}]^\intercal.$$

These functions will have finite means provided $\mathbb{E}[\|X\|^4]$ is finite under $\mu_1$.

## D   Convergence rate when transporting from a uniform distribution

We want to illustrate the convergence rate in Proposition 6.

With the same notations as in problems 1S-MCOT and 1S-RMCOT, we define $\mathcal{X} = [0, 1]$. Distributions $\mu_1$ and $\mu_2$ are the uniform distributions on $\mathcal{X}$. We define $f(x) = x - \alpha$ with $\alpha \in \mathcal{X}$ the imposed mean, and impose a unique constraint: $\langle f, \mu \rangle = 0$.

The cost $c$ is chosen as: $\forall x, y \in \mathcal{X}, c(x, y) = (x - y)^2$.

For these values, it is possible to obtain an explicit solution to 1S-MCOT, using Proposition 3.1:

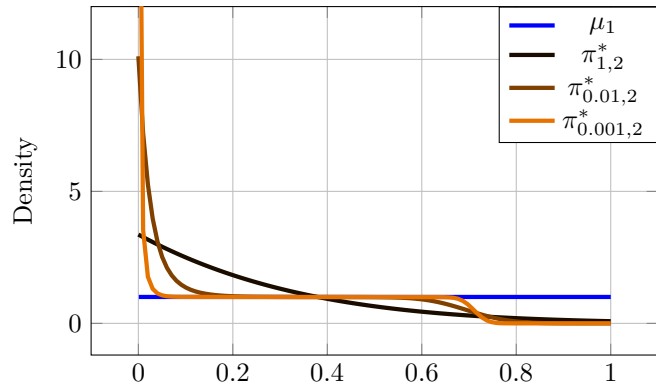

Figure 8: Density of $\mu_1$ and densities of $\pi^*_{\varepsilon,2}$ for different values of $\varepsilon$

$$d^* = \sup_\lambda \int \inf_y [c(x, y) - \lambda f(y)] \mu_1(dx) = \sup_\lambda \int \inf_y [(x - y)^2 - \lambda(y - \alpha)]dx$$

$$= \sup_\lambda \int -\frac{\lambda^2}{4} + \lambda(\alpha - x)dx$$

$$= (\alpha - 0.5)^2.$$

The solution $\pi_\varepsilon^*$ may be obtained through gradient descent as explained in section 3. For $\alpha = 0.25$ and a discretization of $\mathcal{X}$ to 100 points (to compute the gradient), the resulting marginal $\pi_2$ is shown in Fig. 8, achieving the constraint on the mean, for different values of $\varepsilon$.

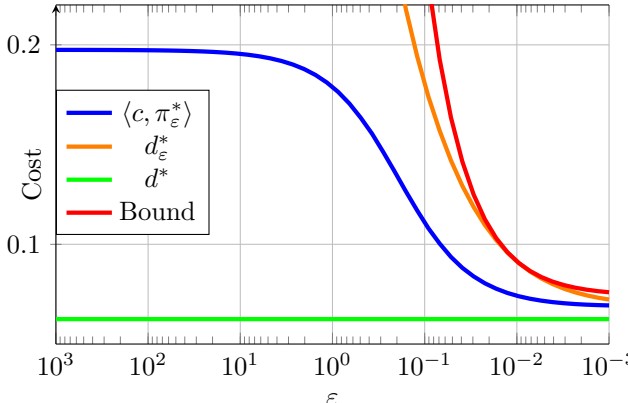

The values of $d^*$ and $\langle c, \pi_\varepsilon^* \rangle$, were obtained for a range of $\varepsilon$ (from $10^{-3}$ to $10^3$). We can observe in Fig. 9 that the convergence to the minimum of the unregularized problem is fast and that it respects the inequality proved in Proposition 6:

Figure 9: Comparison of the costs $d^*$ and $\langle c, \pi_\varepsilon^* \rangle$ for different values of $\varepsilon$

$$|d_\varepsilon^*(\mu_1, \mathcal{P}_f) - d^*(\mu_1, \mathcal{P}_f)| \le \varepsilon D_{\mathrm{KL}}(\pi_0 \| \mu_1 \otimes \mu_2).$$

# E  Costs of the solutions shown in Section 3

In this section, the value of the cost function associated with the distribution $\pi$ produced by the algorithm is reported for the different constraints presented in Section 3. The total cost is decomposed into the contribution of the control cost, $\langle c, \pi \rangle$, and the contribution of the entropic regularization term, $\varepsilon\mathrm{KL}(\pi\|\mu_1 \otimes \mu_2)$, which is intended to remain small.

## E.1  Costs for the solutions shown in Figure 2

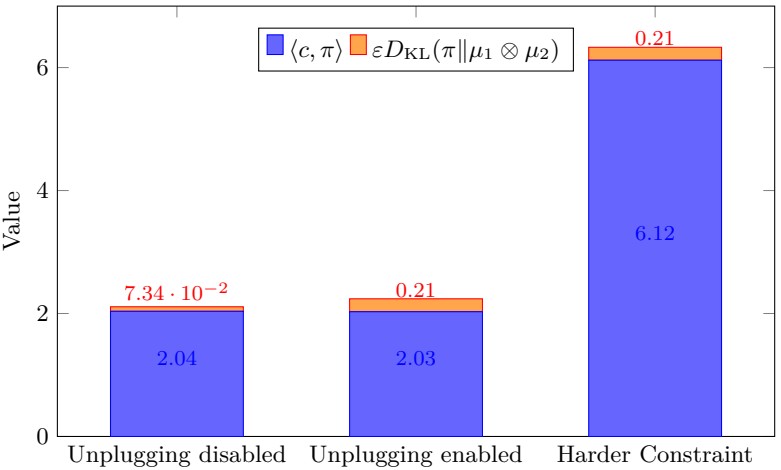

Figure 10: Decomposition of the cost for the different scenarios of Figure 2

Figure 10 shows that allowing vehicles to unplug and replug does not alter the cost $\langle c, \pi \rangle$, as long as the constraint remains easy to satisfy. The cost $\varepsilon KL(\pi^*\|\mu_1 \otimes \mu_2)$ due to the KL divergence cannot be compared between these two scenarios, as the state space is not the same (and thus we do not compare with the same $\mu_2$). The third bar represents a scenario, where unplugging is allowed and an harder constraint is imposed (no consumption during two hours), which results in a significantly higher cost.

## E.2  Costs for the solutions shown in Figure 3

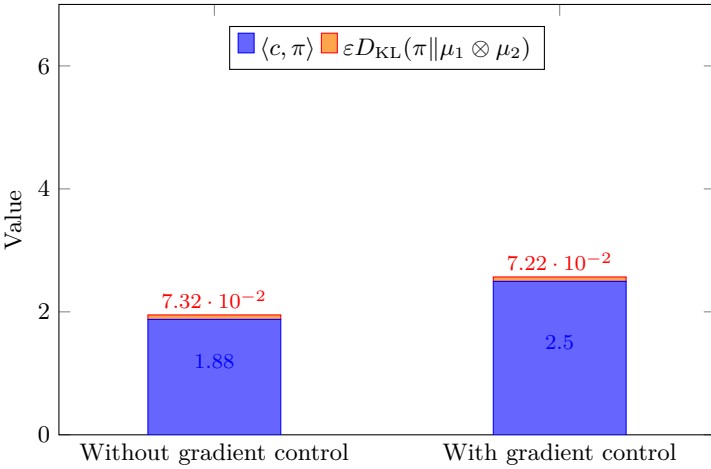

Figure 11: Decomposition of the cost for the two scenarios of Figure 3

Figure 11 shows that adding an additional constraint (here, a constraint on the slope of total consumption) leads to an increased cost. In Figures 10 and 11, the impact of regularization appears to be small, which is consistent with our choice of keeping $\varepsilon$ small.

## F Comparison with methods based on Hamilton-Jacobi-Bellman equations and Piecewise Deterministic Markov Processes

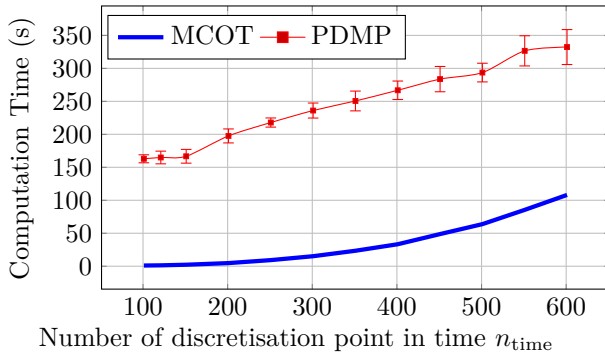

Figure 12: Computation time for MCOT and PDMP (PDMP algorithm is stochastic, error bars are computed over 10 simulations)

A common method in the MFC literature is to start with Hamilton-Jacobi-Bellman equations, discretize these equations and solve them numerically. We compare ourselves here with an article (Séguret et al., 2024) that applies this type of method to EV charging, via a generation of Piecewise Deterministic Markov Processes (PDMP). The case study here is a flat signal, and both methods seek to track this signal. As the PDMP method can only take into account one fixed starting time, we assume that all vehicles arrive at 9am. We note in Figure 12 that our MCOT method is faster in this case, whatever $n_{\text{time}}$. For higher values of $n_{\text{time}}$, MCOT's computation time increases quadratically, and

it could be improved by using Monte Carlo methods to simulate trajectories (as the PDMP method does). Apart from computation time, another advantage of the MCOT method is the flexibility of the model considered: in particular, vehicles arriving at different times can be considered.

## G Water heaters control

We present in this appendix how to apply MCOT-C in a control problem other than the control of EVs.

**Water heater control problem:** We consider a large population of homogeneous Water Heaters (WH). At time t, a WH is modeled by its mean temperature $\theta(t) \in \Theta$, where $\Theta$ is a subset of $\mathbb{R}$, and its power mode $m(t) \in \{0, 1\}$ (Off/On). These WHs follow the Ordinary Differential Equation (ODE):

$$\frac{d\theta(t)}{dt} = -\rho(\theta(t) - \theta_{amb}) + \sigma m(t)p - \sigma\epsilon(t),$$

with $\rho$ the fraction of heat loss by minute, $\sigma$ the specific heat capacity of the volume of water, $p$ the heating power, $\theta_{amb}$ the room temperature, and $\epsilon(t)$ the power equivalent of the water drains at time $t$. Moreover, a water heater aims at keeping its mean temperature between $\theta_{min}$ and $\theta_{max}$ by turning the water heater Off whenever the temperature reaches $\theta_{max}$ and turning it back On whenever the temperature reaches below $\theta_{min}$. The intial density of WHs is at time 0:

$$\theta_0, m_0 \sim \nu_0.$$

And the nominal policy $\mu_1$ is thus defined as follows by this update equation:

$$\begin{cases} \theta_{t+1} = \theta_t - \rho\delta t(\theta_t - \theta_{amb}) + \sigma\delta t m_t p_{max} - \sigma\epsilon_t \\ m_{t+1} = \begin{cases} m_t & \text{if } \theta_{t+1} \in [\theta_{min}, \theta_{max}] \\ 0 & \text{if } \theta_{t+1} \geq \theta_{max} \\ 1 & \text{if } \theta_{t+1} \leq \theta_{min} \end{cases} \\ \theta_0, m_0 \sim \nu_0 \end{cases} \qquad (36)$$

The population of water heaters is represented through its mean-field distribution $\mu$, that is, the empirical distribution obtained in the limit of an infinite number of agents. When applied to a finite but large population, this mean-field distribution induces a control policy. We consider the setting in which a central agent seeks to control these WHs in order to satisfy $A$ global constraints on the aggregate behavior of the WHs:

$$\forall a \in \{1, \ldots, A\}, \ \langle f^{(a)}, \mu \rangle \leq 0,$$

where the function $f$ is defined as $f \colon \mathcal{X} \to \mathbb{R}^A$.

**Application of MCOT-C to this problem:** We allow the WHs to flip their power mode (from On to Off or from Off to On), while the temperature is still between the two bounds $\theta_{t+1} \in [\theta_{min}, \theta_{max}]$. We limit ourselves to two flips per day per water heater. We will note these two times $t_1, t_2 \in \{1, \ldots, T\}^2$. This limitation avoids frequent switching, which is undesirable for the water heater. Therefore, the power mode update for the policy $\mu_2$ can be written as follows:

$$m_{t+1} = m_t \text{ if } \theta_{t+1} \in [\theta_{min}, \theta_{max}] \text{ and } t \notin \{t_1, t_2\}$$
$$m_{t+1} = (1 - m_t) \text{ if } \theta_{t+1} \in [\theta_{min}, \theta_{max}] \text{ and } t \in \{t_1, t_2\}$$
$$m_{t+1} = 0 \text{ if } \theta_{t+1} \geq \theta_{max}$$
$$m_{t+1} = 1 \text{ if } \theta_{t+1} \leq \theta_{min}$$

We can choose $c$ as the number of flips added during the day or the temperature difference at the end of the day, to reduce the impact of the algorithm on user comfort.

**Discussion** This problem can be formulated with the MCOT-C framework:

$$\min_\pi \langle \pi, c \rangle + \varepsilon KL(\pi \| \mu_1 \otimes \mu_2) : \pi \in K(\mu_1), \ \pi_2 \in \mathcal{P}_f$$

An important difference is that here we directly control trajectories (solutions of the ODE). The absolute continuity (imposed by the KL term) ensures that trajectories in $\mu$, the optimal policy, are necessarily solutions of this ODE, as the support of $\mu$ is in the support of $\mu_2$. A second difference is that the state space is very large (it is the number of possible controlled trajectories), and thus the gradient in our algorithm should be computed with Monte Carlo methods. A more detailed discussion of this problem, including a presentation of the Monte Carlo methods, is provided here (Corre et al., 2025).

