# OpenReview forum: "Moment Constrained Optimal Transport for Control Applications"
_TMLR — Accepted by TMLR_

### Review · Reviewer_DLci · 2025-11-22

**Summary Of Contributions:**

The paper considers a moment-constrained optimal transport problem, where one marginal is constrained to "moment classes," which are functional constraints. The authors propose a Sinkhorn-based method to solve the problem, and also provide an online-version of the problem. Experiments are provided on EV charging data.

Strengths:
- The problem formulation seems novel, and the experiments seem quite promising.

Weaknesses:
- The authors do not provide a convergence analysis of the algorithm.
- The presentation of the paper is very much presented with the EV charging application in mind, which may limit perceived applicability for the wider community.

**Additional Comments:**

Typoes and formatting:
- Lasry & Lions (2007) on beginning of page 2 -> use citep
- Please use \eqref for equations
- Litterature -> Literature
- Assumptions paragraph on page 4: \mathbf{R} -> \mathbb{R}
- EV charging with unplugging paragraph on page 7: 10pm -> this should be 10 am?
- Equation 11b, should max power be {1, \dots, n_{power}}?
- Section 4.2: In general, \kappa is chosen... (it is important... during the day). is a runon sentence
- Page 10: I suggest using the term "warm start," e.g. use the previous iteration's output as a warm start for the next iteration

Algorithm 1 and related text:
- K is overloaded for both iterations and the Gibbs kernel;
- inconsistent notation on \zeta_k vs \zeta^k in Algorithm 1 and the whole section

**Audience:**

Yes

**Audience Explanation:**

Control theory and optimal transport have both been more important in applications for machine learning in recent years, so this paper is of interest to the broader community.

**Claims And Evidence:**

Yes

**Claims Explanation:**

Yes, the bulk of the theoretical claims consider the existence of a solution via the dual problem. The results follow from mostly standard arguments from prior work.

**Requested Changes:**

Currently, I'm mostly concerned about the presentation of the work. It is heavily focused on the EV charging example, and I would appreciate it if the authors could add discussion on additional examples where their method could be applicable. It would be even better if they can run experiments on another example. Having additional discussion, and/or a different experiment, would secure my recommendation for acceptance.

Can the authors also provide the cost of the solutions the algorithms produce in Sections 3 and 4? This would help the readers understand how much the additional constraints penalize the cost of the solution, along with another point of comparison against prior work, e.g. in Figure 4.

In the literature section, please explicitly discuss entropic OT, as this method is what machine learning practitioners generally use.

---

> ### Author Response · Authors · 2025-12-13
> **(1) Applicability of the method in other contexts**
>
> We thank the reviewer for their comments, and we will, in particular, take into account the typos and formatting that they pointed out. First, we want to address your concern on the applicability to other examples by providing an example of controlling a large population of water heaters.
>
> **Water heater control problem**
> We consider a large population of homogeneous Water Heaters (WH). At time t, a WH is modeled by its mean temperature $\theta(t)\in \Theta$, where $\Theta$ is a subset of $\mathbb{R}$, and its power mode $m(t)\in \{0,1\}$ (Off/On). These WHs follow the Ordinary Differential Equation (ODE):
> $$
> \frac{d\theta(t)}{dt} = -\rho(\theta(t)-\theta_{amb}) + \sigma m(t)p  - \sigma \epsilon(t),
> $$
> with $\rho$ the fraction of heat loss by minute, $\sigma$ the specific heat capacity of the volume of water, $p$ the heating power, $\theta_{amb}$ the room temperature, and $\epsilon(t)$ the power equivalent of the water drains at time $t$. Moreover, a water heater aims at keeping its mean temperature between $\theta_{min}$ and $\theta_{max}$ by turning the water heater Off whenever the temperature reaches  $\theta_{max}$ and turning it back On whenever the temperature reaches below $\theta_{min}$. The intial density of WHs is at time 0: $$\theta_0,m_0 \sim \nu_0.$$
> And the nominal policy $\mu_1$ is thus defined as follows by this update equation:
>
> $$\theta_{t+1} = \theta_t - \rho\delta t (\theta_t-\theta_{amb}) + \sigma \delta t m_t p_{max} - \sigma \epsilon_t$$
>
> $$m_{t+1} =    m_t \text{ if } \theta_{t+1}\in[\theta_{min},\theta_{max}]$$
>
> $$m_{t+1} =     0 \text{ if } \theta_{t+1} \geq \theta_{max} $$
>
> $$m_{t+1} =     1 \text{ if } \theta_{t+1} \leq \theta_{min} $$
>
> The population of water heaters is represented through its mean-field distribution $\mu$, that is, the empirical distribution obtained in the limit of an infinite number of agents. When applied to a finite but large population, this mean-field distribution induces a control policy. We consider the setting in which a central agent seeks to control these WHs in order to satisfy $A$ global constraints on the aggregate behavior of the WHs:
> $$    \forall a\in\{1,\dots,A\}, \ \langle f^{(a)},\mu\rangle \leq0,
> $$
> where the function $f$ is defined as $f\colon\mathcal{X}\to\mathbb{R}^A$.
>
> **Application of MCOT-C to this problem:** We allow the WHs to flip their power mode (from On to Off or from Off to On), while the temperature is still between the two bounds $\theta_{t+1}\in[\theta_{min},\theta_{max}]$. We limit ourselves to two {flips} per day per water heater. We will note these two times $t_1,t_2\in\{1,\dots,T\}^2$. This limitation avoids frequent switching, which is undesirable for the water heater. Therefore, the power mode update for the policy $\mu_2$ can be written as follows:
>
> $$m_{t+1} = m_t  \text{ if } \theta_{t+1}\in[\theta_{min},\theta_{max}] \text{ and } t\notin\{t_1,t_2\}$$
> $$m_{t+1} =  (1-m_t)  \text{ if } \theta_{t+1}\in[\theta_{min},\theta_{max}] \text{ and } t\in\{t_1,t_2\}$$
> $$m_{t+1} = 0   \text{ if } \theta_{t+1} \geq \theta_{max} $$
> $$m_{t+1} = 1 \text{ if } \theta_{t+1} \leq \theta_{min} $$
>
> We can choose $c$ as the number of {flips} added during the day or the temperature difference at the end of the day, to reduce the impact of the algorithm on user comfort.
>
> **Discussion** This problem can be formulated with the MCOT-C framework:
> $$\min\limits_{\pi}  \\{ \langle \pi , c \rangle + \varepsilon KL(\pi\|\mu_1\otimes\mu_2) :   \pi \in  K(\mu_1) ,  \  \pi_2 \in \mathcal{P}_f \\} $$
>
> An important difference is that here we directly control trajectories (solutions of the ODE). The absolute continuity (imposed by the KL term) ensures that trajectories in $\mu$, the optimal policy, are necessarily solutions of this ODE, as the support of $\mu$ is in the support of $\mu_2$. A second difference is that the state space is very large (it is the number of possible controlled trajectories), and thus the gradient in our algorithm should be computed with Monte Carlo methods.

---

> > ### Author Response · Authors · 2025-12-13
> > **(2) Applicability of the method in other contexts**
> >
> > The current presentation aims to illustrate the MCOT-C framework using the example of EV charging, in order to clarify and make it more understandable. It does not emphasize the applicability to other examples, so we will add a discussion in the conclusion on the subject of controlling trajectories:
> >
> > "One-sided moment relaxation of OT problem provides a very natural representation setting for mean field control applications. This framework considers problems where the initial distribution is known and the goal is to reach a final distribution that satisfies moment constraints, while minimizing a certain control cost. A direct application is found in electric vehicle charging, where the objective is to optimally schedule charging to control their aggregate consumption. It could also be applied to more complex problems in which the distributions represent distributions of trajectories that are solutions of differential equations, for instance, the temperature evolution of a water heater. In such contexts, MCOT-C ensures that trajectories in the optimal policy $\mu$ are necessarily solutions of this ODE, through absolute continuity. Beyond demand response, other potential applications can be envisioned, such as controlling a population of drones to provide flexible network coverage services [1]."
> >
> > [1] Chen, Dezhi, et al. "Mean field deep reinforcement learning for fair and efficient UAV control." IEEE Internet of Things Journal 8.2 (2020): 813-828.

---

> > > ### Comment · Reviewer_DLci · 2025-12-19
> > >
> > > Thanks for the clarifications. This addition in the conclusion, along with adding this water heater control scenario for instance to the appendix, would be helpful for practitioners.

---

> > > > ### Author Response · Authors · 2025-12-24
> > > >
> > > > We have just added the water heater control scenario in the appendix of the new version of the pdf.

---

> ### Author Response · Authors · 2025-12-13
> **(3) Cost of the solution and Litterature**
>
> **Cost of the solution in the numerical example**
>
> Regarding the cost of the solutions, we will include a comparison in the appendix between the costs for each type of constraint (Fig. 2 and 3 in the article). The figures are available here: https://anonymous.4open.science/r/TMLR_SolutionCost-FFB7/TMLR_CostSolutions.pdf
>
> What the first figure shows is that allowing vehicles to disconnect and reconnect does not alter the cost $\langle c,\pi\rangle$ when comparing the first and second bars, as long as the constraint remains easy to satisfy. The cost $\varepsilon KL(\pi^*\|\mu_1\otimes\mu_2)$ due to the KL divergence cannot be compared between these two scenarios, as the state space is modified (and thus we do not compare with the same $\mu_2$). The third bar represents a scenario with a much stricter constraint, which results in a significantly higher cost. In the second figure, we observe that adding an additional constraint (here, a constraint on the slope of total consumption) leads to an increased cost. Across all these cost values, the impact of regularization appears to be minimal, which is consistent with our choice of keeping $\varepsilon$ small.
>
> **Literature**
> We will modify the literature section to add a discussion on entropic OT:
>
> "The introduction of an entropic regularizer, which leads to solutions that are easily computable, has become standard in OT since [2]. This development led to the entropic optimal transport problem, which is closely related to the one considered here (except that the constraint on the second marginal is replaced by a moment constraint) both in its formulation and in the algorithms used to solve it."
>
> [2] Cuturi, "Sinkhorn distances: Lightspeed computation of optimal transport." Advances in neural information processing systems 26 (2013).

---

### Review · Reviewer_Hpy6 · 2025-11-29

**Summary Of Contributions:**

This paper proposes a mean-field control framework based on ideas from optimal transport (OT) for coordinating large populations of controlled agents. The key technical step is to relax the standard OT formulation to a one-sided problem: the first marginal (current empirical distribution of agents) is fixed, while the second marginal is constrained to lie in a “moment class,” i.e., a set of probability measures satisfying generalized moment constraints. This lets the controller specify aggregate constraints (moments) instead of a full target distribution. The authors further introduce an entropic regularizer, which both improves computational tractability and allows them to encode hard constraints on individual agent behavior. They derive a Sinkhorn-inspired algorithm to solve the resulting problem efficiently.

To demonstrate the approach, they apply it to distributed control of electric-vehicle charging under grid constraints. They present both an offline formulation and an online version, and validate the method on an EV dataset.

**Audience:**

No

**Audience Explanation:**

I am actually not sure it fits the scope of the journal. I do not have a strong opinion. I suppose it could fit the scope "development of new analytical frameworks that advance theoretical studies of practical learning methods", but I am not sure.

**Broader Impact Concerns:**

None.

**Claims And Evidence:**

No

**Claims Explanation:**

The authors should address the numerous comments I collected below.

**Requested Changes:**

Critical to securing recommendation for acceptance:
1. The paper is *riddled* with typos or missing acronyms. It is the authors responsibility to perform careful proofread of their own work. Examples (definitely non-exhaustive):
   -  page 1 (abstract): EV acronymin in abstract is undefined: you do introduce the acronym in the introduction… i would suggest not to use the acronym here.
   -  page 2: "Litterature"
   -  page 3: "MFC" (I suppose mean-field control). Also in general, you should be consistent and once you introduce an acronym, use it
   -  page 4: equation 4, missing "d" in the integral
   - ....
   - and continues on the appendix: "Eq. equation 7", the lambda with ^1 bold or not bold is too similar and confusing, etc.
2. page 1 (and others): The function f used for the constraints is sometimes used as a vector-valued function, but then integrals and inequalities are used without any clear meaning. It is unclear whether they mean multiple constraints (and with which math notation?), or a single constraint. Please fix. Be also consistent with the lagrange multipliers in appendix A.
3. page 1: "Here, Sk denotes an exogenous variable, while Wk is a fully controllable decision variable." Sounds as if S_k is imposed onto the agent (e.g., a disturbance) - should be endogenous? and Wk is only a control input (what does it mean fully controllable decision variable? a decision variable is what you pick… did you mean fully controllable endogeneous variable?) I think generally this problem setup is not very clear, and should be exemplified.
4. page 1: "Sk is the EV’s arrival time at the parking lot and its initial state of charge". If you are solving one instance of such problem, the arrival time does not seem something that needs to be introduced in the state but rather a parameter of the problem. related to my other comment above, it is unclear to me why the state of charge would be exogenous.
5. page 1: what is a vehicle transaction?
6. page 2: "[...] it is often impractical [...]" This is not at all obvious to me. You could make it concreet by eg explaining what the dynamics are, the costs… is it not a convex problem? … what optimization approaches can be introduced? what is the solving time? this context would be helpful.
7. page 2: You mention this "exogenous" variable again. I suppose what you are saying here is that you are steering a distribution of agents from an initial configuration to a final one: why not say so? I find this exposition rather convoluted. Please simplify your exposition, and clarify your problem setting. My recommendation is to have a clear example (small scale!) in which you can name and define mathematically all variables explicitly, to make extra clear to the reader what is the setting.
8. page 2: "In the present work, we consider a slightly modified formulation in which only the final distribution is required to satisfy the moment constraints. We show, for the first time, that this formulation can be interpreted as a mean-field control problem." This seems wrong. While novelty is not important, you should be accurate in your claims. How can your problem setup be a special case yet be the first time to show a property that was shown for the general case? Elaborate.
9. page 2 (contributions): What is a "strong" constraint?
10. page 2 (contributions): Under the two conditions mentioned in the first bullet point, the charging problem is only a scheduling problem.
11. page 2 (contributions): "We propose an algorithm to solve MCOT-C, with a Sinkhorn update on one side and a gradient descend update on the second side." This is basically a special case of Frank-Wolfe, please compare appropriately with this, existing literature etc. In general, the literature is barely acknowledged. I am not concerned about claims of novelty, but I cannot appreciate the work or see why it should be published if the authors do not clearly contextualize their work.
12. page 2 (contributions): "We extend this approach to an online setting, where the data about the EVs is progressively discovered and show its efficiency on a case study with a real data set (OpenDataset, 2019)." I do not see the "efficiency" of the proposed method from the experiments in section 4. Please elaborate.
13. page 2 (contributions): "Compared with the existing literature, our model allows to take into account new global constraints (slope of global vehicle consumption, for example)" Your comparison with the literature is limited. First, it is not obvious why a proper solver would not be able to handle this. Second, there is existing work that would allow to study necessary conditions for the problem (they are rather trivial in your problem setting), and I would not be surprised if they yielded easy learning problems. See e.g.  https://arxiv.org/abs/2406.10676. Can you please compare with these and other works and explain how previous works do not allow to tackle the problem or do so inefficiently? Your literature review focuses on why your method is new, but not why it is needed. There are arbitrarily many new methods that one can come up with.
14. page 2 (contributions): "to obtain faster results, especially when the size of the control space is small, and to take into account predictions about the future in the form of probability densities." If the size of the control space is small, why would you need a mean field formulation (note that in your formulation the control is roughly the fleet size)? Also, your "faster" is misleading, you tune the speed of your algorithm with epsilon in the regularizer, which is basically changing the problem solution (see comments below).
15. page 2 (contributions): "and to take into account predictions about the future in the form of
probability densities." If all the methods solve 2, I do not see why this observation matters. It is either done explicitly or implicitly. Please elaborate.
16. page 4: There is a catch: Each regularization yields a different solution. Please comment on that, what are the pitfalls or why it does not matter in your case. (if that is the case....)
17. page 4 (assumptions): "Assumptions are required for the existence of optimizers and desirable properties of the dual" This is not how assumptions should be stated or formulated, in particular for applied work. First, required is not the right word: You do not prove necessity of these assumptions, only sufficiency. Or do you? Second, you should connect the assumptions with the problem of interest. What are the limitations of your assumptions?
18. page 4 (assumptions): "have compact support" how is this compatible with what you report in appendix C?
19. page 4 (assumptions): "is positive definite" when is this not true? discuss
20. page 6: "The goal is power tracking: total power consumption should follow a reference signal (rt) over a time period" elaborate why this is relevant in practice.
21. page 6: This problem setup (both with plugging and unplugging) is a simple static optimization problem as formulated. You should compare with a solver...
22. page 6:  You talk about complexity. Please be precise about what do you mean by "complexity". Number of operations? Is it best case, worst case, average case?
23. page 6: When you select epsilon, you should also clarify how this changes the optimal solution.
24. page 7: "is extended with two extra time dimensions" why only two? can’t they unplug multiple times?
25. page 7-8: "Faster results than HJB methods" You need to report how well the solution matches, otherwise this section brings little to no information. It would also be interesting to provide a comparison with other methods, like a global solver and an analytical solution (see my other comments in introduction).
26. page 8: "Apart from computation time, another advantage ..." Do you show this? Not clear, please elaborate.
27. page 12: "leads to a tractable algorithm by directly considering only the distribution moments that are relevant for control." Inaccurate. You still introduce approximations of other kind, you also use a KL divergence and not solve the same problem. And below you also talk about the fact that you need to discretize…
28. page 12: "advanced optimization techniques" These are not advanced, they are in fact fairly standard.
29. page 12: "infinite" it is rather unusual to have moment constraints on distributions with finite, discrete support..
30. page 12: "As the complexity of the algorithm increases with the size of this state space, it may be necessary to adapt this method to limit computation time, by using Monte Carlo-type methods, i.e. generating a number of trajectories to obtain an approximation of the gradient, instead of calculating it exactly." seems a major limitation. Also, it is never clear in the text that in the end you end up doing this until very late, you should very clearly specify so in the introduction.
31. page 12: "We believe that representing distributions by their moments to perform optimal transport has broader applications in machine learning and control. We aim to explore its potential in other contexts." this is too broad and uninformed. please be more concrete or remove.
32. page 15: "Compactness is assumed in ..." You may want to point to work going beyond compactness. See eg https://arxiv.org/abs/2411.02549 for a review. Also, compactness for canonical distributions is unusual. What are canonical distributions? Gaussians? Not compactly supported....
33. page 15: Report proofs right below the statements in an appendix.. Also make sure that if you state a proposition, you report the proof fully. Here the devil is in the detail. Do not use handwavy arguments like "is based on approximation". Show the argument precisely.
34. page 15: Now the sum over the policy is on a discrete set... Very confusing.
35. page 21: The exposition needs to be improved. E.g., "Computation for non-Gaussian" comes out of the blue. I am not sure how this fit in the text…. is it part of the proposition? Is it not? The authors should somewhat discuss and guide the reader through the section.
36. page 22: I do not understand the plots. Please write the axis labels properly. I do not see the need to save space in an appendix.
37. page 22: The ranges of epsilon make no sense. You are solving completely different problems, what is the point of comparing the convergence speed? Show also the difference in the solution... or elaborate.

Others:
1. page 1: "We adopt a distributed control framework". I would recommend to write it the other way around: First, you say the problem you want to solve. then, you say that you can tackle it with a distributed framework.
2. page 2 (contributions) and everywhere: When you have the list (i), (ii), .. within a sentence, the punctuation should be appropriate. Your capitalization and punctuation puzzles me and makes it hard to read the paragraph. Similarly all around equations your punctuation is wrong. Equations also should have punctuation if eg you start a new sentence...
3. page 3 (and in other parts): "where the second marginal is not fully specified but must belong to a set of distributions respecting moment constraints." this is not the only information. You also know they are cost minimizing.
4. page 21: Proof of proposition 9. Is it not well known? Please check carefully the literature for your results and where possible cite prior work rather than proving the result yourself.

I am not checking the proofs in detail until the authors have addressed the above comments.

---

> ### Author Response · Authors · 2025-12-15
> **Responses to remarks 1 - 7**
>
> We thank the reviewer for their careful and detailed comments, which we address individually below.
>
> **1.**  We appreciate the reviewer’s observation and will correct the typographical errors and abbreviations accordingly.
>
> **2.**  *page 1 (and others): The function $f$ used for the constraints is sometimes used as a vector-valued function, but then integrals and inequalities are used without any clear meaning. It is unclear whether they mean multiple constraints (and with which math notation?), or a single constraint. Please fix. Be also consistent with the lagrange multipliers in appendix A.*
>
> The function $f$ is a vector-valued function defined as $f:\mathcal{X}\to\mathbb{R}^M$. The inequality $f(x)\leq 0$ is meant componentwise, i.e., $\forall m \in\{1,...,M\}, f^{(m)}(x)\leq 0$, where $f^{(m)}(x)$ is the m-th component of the vector $f(x)$. We will clarify this notation when $f$ is introduced on page 1.
>
> **3.** *page 1: "Here, Sk denotes an exogenous variable, while Wk is a fully controllable decision variable." Sounds as if Sk is imposed onto the agent (e.g., a disturbance) - should be endogenous? and Wk is only a control input (what does it mean fully controllable decision variable? a decision variable is what you pick… did you mean fully controllable endogeneous variable?) I think generally this problem setup is not very clear, and should be exemplified.*
>
> The wording "exogenous" and “fully controllable decision variable” has been removed for clarity, and we will change the problem setup to "Here, $S_k$ denotes a non-controllable variable, while $W_k$ is a control (or decision) variable.".
>
> **4.** *page 1: "Sk is the EV’s arrival time at the parking lot and its initial state of charge". If you are solving one instance of such problem, the arrival time does not seem something that needs to be introduced in the state but rather a parameter of the problem. related to my other comment above, it is unclear to me why the state of charge would be exogenous.*
>
> In the EV charging example,  indeed $S_k$ is the arrival time and the state of charge at the arrival (which cannot be controlled by the central planner) and $W_k$ is the plugging time (which is the control variable in this example).
> While we agree that for one instance of the problem the initial arrival time and state of charge can be used as a parameter, we will be considering a large population of agents, represented by a distribution of the initial state at the limit when the size of the population goes to infinity. By omitting the initial state, the exposition for the mean-field control setting becomes less practical. Furthermore, more evolved examples may contain as state description the trajectories of agents over the whole time horizon, see also response (1) to reviewer **DLci** containing an example of controlling the power consumption of a large population of water-heaters.
>
> **5.**  *page 1: what is a vehicle transaction?*
>
> The term is not clear in this context, and we will change it to "charging sessions".
>
> **6.** *page 2: "[...] it is often impractical [...]" This is not at all obvious to me. You could make it concreet by eg explaining what the dynamics are, the costs… is it not a convex problem? … what optimization approaches can be introduced? what is the solving time? this context would be helpful.*
>
> Here, the coupling due to the constraint $\sum_{k=1}^K f(X_k) \leq 0$ implies that we cannot control each agent separately. In the EV example, it is not possible to optimize for each EV separately, as they all impact total power consumption. Therefore, the control parameter for $N$ agents is the vector of all their plugging times, and optimizing over this vector becomes impractical when $N$ is very large (e.g. 10 000 or more). With mean field methods, the control parameter is the distribution of the plugging times and so the complexity due to $N$ vanishes.
>
> **7.** *page 2: You mention this "exogenous" variable again. I suppose what you are saying here is that you are steering a distribution of agents from an initial configuration to a final one: why not say so? I find this exposition rather convoluted. Please simplify your exposition, and clarify your problem setting. My recommendation is to have a clear example (small scale!) in which you can name and define mathematically all variables explicitly, to make extra clear to the reader what is the setting.*
>
> When replacing the term exogenous with non-controllable, we hope that it will be clearer for the reader. We were already saying that we are stearing a distribution of agents from an initial configuration to a final one in the paragraph after equation (2).  We will also refer in the introduction to the "Use Case" section, which formally introduces the problem formulation in a concrete small scale (the control variable is one dimensional) context. Also, a more evolved example is provided in response (1) to reviewer **DLci**.

---

> > ### Comment · Reviewer_Hpy6 · 2025-12-16
> > **Responses to remarks 1 - 7**
> >
> > 1. There are still issues:
> > - Now acronyms appear introduced multiple times in the text (e.g., OT) and after being introduced they are not consistently used (e.g., MFC; why not using the glossary package in latex?)
> > - "Notation", not "Notations"
> > - ...
> >
> > 2-5. Thanks!
> >
> > 6. If it is a linear program I do not see an issue with N = 10000. I think you can clarify why this is a hard problem (if it is).
> >
> > 7. I still think you could make the exposition more pedagogical with a concrete very small example in which you can write the formulas for the distributions involved.

---

> ### Author Response · Authors · 2025-12-15
> **Contributions (remarks 8 - 15)**
>
> **8.** *page 2: "In the present work, [...] We show, for the first time, that this formulation can be interpreted as a mean-field control problem." This seems wrong. [...]*
>
> Thank you for pointing this out, indeed the statement was very confusing, and not highlighting well our contribution.
> In the problem introduced by Alfonsi et al, MCOT is proposed as an approximation method to solve a classical OT problem. The novelty of our work is that we propose a new approach for modeling mean-field control problems as a one-sided MCOT variant.
> Although this change is minimal mathematically speaking, this one-sided variant is particularly well-suited for mean-field control. To be more precise, we replaced: "We show, for the first time, that this formulation can be interpreted as a mean-field control problem." by
> "The novelty of our work is that
> we propose a new approach for modeling MFC problems as a one-sided MCOT variant, in which only the final distribution is required to satisfy the moment constraints. This one-sided variant is particularly well-suited for MFC, where the initial distribution corresponds to the nominal dynamics of the system, while the target one is only specified through the moment constraints. For example, in the demand response applications in power grids, the coordinator only cares about the constraints on the power consumption of the whole population of flexible devices, and not about the detailed distribution over all individual device states."
>
> **9.** *page 2 (contributions): What is a "strong" constraint?*
>
> It is a typo, thank you for pointing it out, the correct term is "hard constraint".
>
> **10.** page 2 (contributions): Under the two conditions mentioned in the first bullet point, the charging problem is only a scheduling problem.
>
> Yes, we could also consider this problem from a scheduling point of view. A comparison with a scheduling algorithm from the literature is proposed in Section 4.7.
>
> **11.** *page 2 (contributions): "We propose an algorithm to solve MCOT-C, with a Sinkhorn update on one side and a gradient descend update on the second side." This is basically a special case of Frank-Wolfe, [...]*
>
> Our algorithm is a projected gradient descent algorithm. The fact that it looks like the Sinkhorn algorithm is a remark, and we will modify the contribution to: "We propose a projected gradient descent algorithm to solve MCOT-C and highlight its similarity to the Sinkhorn algorithm."
>
> **12.** *page 2 (contributions): "We extend this approach to an online setting, where the data about the EVs is progressively discovered and show its efficiency [...]*
>
> In Section 4, we show that we are able to apply our approach on a real dataset, and we compare our results with another algorithm. We will replace the term "efficiency" with "applicability" to make it clearer.
>
> **13.** *page 2 (contributions): "Compared with the existing literature, our model allows to take into account new global constraints (slope of global vehicle consumption, for example)" Your comparison with the literature is limited. [...]*
>
> We compare our work with respect to the literature on mean-field control problems involving a large population of flexible electricity consumers. Within this framework, and to the best of our knowledge, the constraints considered in our model are new. This contribution is intended to clarify our positioning with respect to this part of the literature, and we will make this explicit by adding: "Compared with the existing literature on mean-field control for demand response".
>
> **14** *page 2 (contributions): "to obtain faster results, especially when the size of the control space is small, and to take into account predictions about the future in the form of probability densities." If the size of the control space is small, why would you need a mean field formulation (note that in your formulation the control is roughly the fleet size)? [...]*
>
> Here, the control space is $\mathcal{W}$. In the electric vehicle charging example, it is for instance $[0,T]$, corresponding to a single plugging time. Bigger control spaces can be considered, as in the example with unplugging and re-plugging (Section 3.2), where $\mathcal{W} = [0,T]^3$ (plugging, unplugging, and re-plugging time). The "size of the control space" therefore refers to the dimension of $\mathcal{W}$, and not to the fleet size. We removed the expression "especially when the size of the control space is small", which may be missleading.
>
> **15.** *page 2 (contributions): "and to take into account predictions about the future in the form of probability densities." If all the methods solve 2, I do not see why this observation matters. It is either done explicitly or implicitly. Please elaborate.*
>
> This is correct, other methods achieve this (at least implicitly). We will therefore remove this part of the contributions, as it may be misleading.

---

> > ### Author Response · Authors · 2025-12-16
> > **Answers to remarks 16 - 24**
> >
> > **16.** *page 4: There is a catch: Each regularization yields a different solution. Please comment on that, what are the pitfalls or why it does not matter in your case. (if that is the case....)*
> >
> > In the appendix, we show theoretically and on an example, the convergence of the regularized problem to the unregularized problem, when $\varepsilon$ goes to $0$. Thus, in mean field control use cases, we choose $\varepsilon$ low in order to limit the impact of the regularizer.
> > In applications to power grids, the main goal is to satisfy the moment constraints (e.g. the overall power consumption should not go above a given limit), while the cost $c$ captures the individual effort of users. This information can then be used, for instance, to compute the compensation that users should receive for their flexibility. The aggregator that is solving the problem is therefore left with a tradeoff between easier problem to solve (for higher values of epsilon), vs. less costly solutions but that take more time to compute (for small epsilon.
> > Note also that $\apsilon$ should not be set to zero for mean-field control applications, as it has a dual role: the density $\mu_2$ also encodes hard constraints on the controlled dynamics via the absolute continuity of KL.
> >
> > **17** *page 4 (assumptions): "Assumptions are required for the existence of optimizers and desirable properties of the dual" This is not how assumptions should be stated or formulated, [...]*
> >
> > We will add a discussion on the assumptions in the revision.
> >
> > **18** *page 4 (assumptions): "have compact support" how is this compatible with what you report in appendix C?*
> >
> > Appendix C does not make this assumption of compactness (it is not the case for gaussian variables). For the EV charging example, the state space $\mathcal{X}$ is a product of closed and bounded subsets of $\mathbb{R}$ and thus compact.
> >
> > **19.** *page 4 (assumptions): "is positive definite" when is this not true? discuss*
> >
> > It is not true, if for $Y \sim \mu_2$, $Y$ has linear dependence. From the EV example perspective, it implies that each variable in $Y$ is not a linear combination of the others. If this is the case, it is easy to change $Y$ by removing this variable.
> >
> > **20.** *page 6: "The goal is power tracking: total power consumption should follow a reference signal (rt) over a time period" elaborate why this is relevant in practice.*
> >
> > In power grids, the power production and demand have to be balanced at all times. If this is not the case, the grid frequency changes, which can lead to cascading effects and eventually a blackout. A grid operator generates a (frequency) regulation signal and balancing resources (in the past power generators) are asked to respond to this signal in real time (by ramping up or down their production). Following this regulation signal by flexible demand would limit the use of fossil energy for this balancing ancillary service for the power grid operators. For a longer discussion on the topic see for instance:
> > - https://ieeexplore.ieee.org/document/9721269
> >
> > **21.** *page 6: This problem setup (both with plugging and unplugging) is a simple static optimization problem as formulated. You should compare with a solver...*
> >
> > Section 3 is only a toy example to illustrate the approach. We only report in this section the mean-field limit, i.e. when the number of agents is infinite. Section 4 will consieder an adaptation to a finite population size, revealed online.
> >
> > **22.** *page 6: You talk about complexity. Please be precise about what do you mean by "complexity". Number of operations? Is it best case, worst case, average case?*
> >
> > Here, the complexity is the number of operations to compute the gradient at each iteration. It is computed through matrix multiplication. It is worst case complexity.
> >
> > **23.**  *page 6: When you select epsilon, you should also clarify how this changes the optimal solution.*
> >
> > It is difficult to quantify the impact of $\varepsilon$ in this use case. For this reason, we prefer to discuss this impact theoretically in the appendix and empirically on a simple example in Appendix D. To provide some intuition, we will add this sentence on page 6: "This regularizer penalizes the entropic discrepancy between $\pi$ and $\mu_1 \otimes \mu_2$. As a result, it pushes the optimal policy $\pi_2$ to remain close to $\mu_2$, which will later be chosen as a uniform distribution."
> >
> > **24.** *page 7: "is extended with two extra time dimensions" why only two? can’t they unplug multiple times?*
> >
> > The goal of this section is to provide a toy example (EV charging without unplugging) and to show how to complexify by adding control dimensions (EV charging with unplugging). For multiple plugging times, we just need to add the corresponding dimensions. While it becomes more difficult to compute the gradient in reasonable time with matrix multiplication, it would also be possible to compute it via Monte Carlo methods, to obtain a stochastic gradient descent algorithm.

---

> > > ### Author Response · Authors · 2025-12-16
> > > **Answers to remarks 25 - 37**
> > >
> > > **25.** *page 7-8: "Faster results than HJB methods" You need to report how well the solution matches, otherwise this section brings little to no information. It would also be interesting to provide a comparison with other methods, like a global solver and an analytical solution (see my other comments in introduction).*
> > >
> > > The PDMP-based method controls the jump intensity between modes (ON/OFF), which is not the case in our approach. Therefore, the two methods can only be compared in terms of their compliance with the total consumption constraints. In this example, since the constraint corresponds to a flat signal, the graph shows that both consumption profiles satisfy this constraint, providing limited additional insight.
> > >
> > > **26.**
> > > *page 8: "Apart from computation time, another advantage ..." Do you show this? Not clear, please elaborate.*
> > >
> > > Compared to this method on this use case, the computation time is empirically faster as shown in Figure 4. In general, the resolution of the discretized HJB equations and the generation of a large number of trajectories to obtain their result is computationally expensive.
> > >
> > > **27.**
> > > *page 12: "leads to a tractable algorithm by directly considering only the distribution moments {...]*
> > >
> > > This sentence is not clear. We will rephrase it as:
> > > "By introducing an entropic regularization which allows for obtaining an explicit expression of the gradient, MCOT-C leads to a tractable algorithm."
> > >
> > > **28** *page 12: "advanced optimization techniques" These are not advanced, they are in fact fairly standard.*
> > >
> > > We agree and remove the word "advanced".
> > >
> > > **29** *page 12: "infinite" it is rather unusual to have moment constraints on distributions with finite, discrete support..*
> > >
> > > The term infinite is ambiguous here, as $\mathcal{X}$ is already infinite in the EV example, so we will remove it. What we mean is that in the EV example, we solve the problem by discretizing the time and state space with $N_t$ and $N_s$ points. In some examples, such as the control of water heaters, where the temperature is the solution of a differential equation, we may not want to discretize the state space.
> > >
> > > **30.** *page 12: "As the complexity of the algorithm increases with the size of this state space, it may be necessary to adapt this method to limit computation time, by using Monte Carlo-type methods, [...]
> > >
> > > In this article, we don't rely on Monte Carlo-type methods and only use matrix multiplication to compute the gradient. The discussion on Monte Carlo methods concerns possible extensions of this work. See also discussion in response (1) to reviewer DLci.
> > >
> > > **32.**  *page 15: "Compactness is assumed in ..." You may want to point to work going beyond compactness. See eg https://arxiv.org/abs/2411.02549 for a review. Also, compactness for canonical distributions is unusual. What are canonical distributions? Gaussians? Not compactly supported....*
> > >
> > > We removed "canonical distributions" as this is indeed misleading.
> > >
> > > **33.** *page 15: Report proofs right below the statements in an appendix.. Also make sure that if you state a proposition, you report the proof fully. Here the devil is in the detail. Do not use handwavy arguments like "is based on approximation". Show the argument precisely.*
> > >
> > > We will do our best effort to make the material in Appendix A and B more linear. However, we prefer to keep some technical lemmas in Appendix B to keep the exposition more fluid. However, we will recall the statement of the lemma before its proof.
> > >
> > >
> > > **34.** *page 15: Now the sum over the policy is on a discrete set... Very confusing.*
> > >
> > > It is a typo, we will change it to an integral.
> > >
> > >
> > > **35.** *page 21: The exposition needs to be improved. E.g., "Computation for non-Gaussian" comes out of the blue. I am not sure how this fit in the text…. is it part of the proposition? Is it not? The authors should somewhat discuss and guide the reader through the section.*
> > >
> > > We will change it as a remark.
> > >
> > > **36.**  *page 22: I do not understand the plots. Please write the axis labels properly. I do not see the need to save space in an appendix.*
> > >
> > > We added the axis labels. For Fig 9, the y-axis is the value of the density of each distribution. For Fig 10, the y-axis is the value of the costs and x-axis is the value of $\varepsilon$.
> > >
> > > **37.** *page 22: The ranges of epsilon make no sense. You are solving completely different problems, what is the point of comparing the convergence speed? Show also the difference in the solution... or elaborate.*
> > >
> > > Fig 10 shows the convergence of the regularized problem to the unregularized problem when $\varepsilon$ goes to $0$. As we are interested in the solution to the unregularized problem, we wanted to look at this convergence. We will add in Fig 9, $\pi^*_{\varepsilon,2}$ for other values of $\varepsilon$ across this range.

---

> > > > ### Comment · Reviewer_Hpy6 · 2025-12-16
> > > > **Answers to remarks 25 - 37**
> > > >
> > > > 25-28. Thanks!
> > > >
> > > > 29. How do you do then when you do not discretize the solution? Can you add this discussion?
> > > >
> > > > 30-37. Thanks!

---

> > > ### Comment · Reviewer_Hpy6 · 2025-12-16
> > > **Answers to remarks 16 - 24**
> > >
> > > 16. I think you could make this more explicit. Even in the contributions: e.g., "Drawing ideas from regularized optimal transport, we introduce and tunable entropic regularization that allows to ...". So that it is explicit that you are changing the solution but you provide other benefits.
> > >
> > > 17. "The following assumptions are introduced for the existence of optimizers and desirable properties of the dual." I would remove this sentence, and say that throughout this work you will consider the following assumptions. After the assumptions, you can say that as you will show, these assumptions are sufficient to guarantee the existence of optimizers ... The issue I have with the sentence is that: 1. seems to motivate the assumptions by theoretical properties rather than practical instances, and 2. sounds like they are necessary assumptions (they are only sufficient).
> > >
> > > 18. I find confusing, then, why even bothering mentioning the Gaussian case. It is not used in the application and is not part of the main theory. If you really want to mention it, why not explicitly putting it as a remark? "We relax the assumptions of compactness by considering gaussian distr ..  in ....". Also
> > >
> > > 19-20. Ok, please add these comments in the paper.
> > >
> > > 21. Ok
> > >
> > > 22. Why not using the O notation as in other parts of the paper?
> > >
> > > 23. Figure 9 shows d_\epsilon^\ast \approx 4 times d^\ast. It does not look like it is converging. Perhaps you may want to run the algorithm for epsilon down to 10-6.
> > >
> > > 24. I would mention this as a limitation. Or you could discuss that in practice a finite number of unplugging times is necessary, or perhaps show how the solution changes empirically.

---

> > ### Comment · Reviewer_Hpy6 · 2025-12-16
> > **Contributions (remarks 8 - 15)**
> >
> > 8. Thanks
> > 9. Thanks
> > 10. I think in general what is hard to parse to appreciate the use case EV charging is a "clean" formulation in which, using the exact same notation as in equations (1)-(2) you introduce the problem. I can read c, but it is unclear what f is explicitly. Seems like you are putting a lot of emphasis instead on the KL term, which in principle is only a computational tool.
> > - The advantage seems minimal from the plot and numbers (~1%?). Perhaps you could spell out to the reader why this 1% matters.
> > - How do the results change for different initial conditions?
> > - Complexity in practice depends on the constants as well. What is the execution time of each algorithm?
> > - How does the result depend on the regularization epsilon?
> > - 4.1 Item 5, you discuss this f which I am not sure is the same f as in (1)-(2). Is it? I am a bit puzzled about the notation. I think the issue is that you are introducing a time-varying f I suppose, and the indexing becomes confusing. Can you please clarify? Also sometimes you write f_t, sometimes f^(t).
> >
> > 11-15. Thanks!

---

> ### Author Response · Authors · 2025-12-24
> **Responses to remarks 1-29**
>
> **1.** We will make a revision to take into account these issues.
>
> **6.** For the N-agent problem, it is not a linear problem. If we consider the EV use case, the variable we want to optimize on is the vector of all the plugging times, and the cost $\langle c,\pi \rangle$ is quadratic with respect to these plugging times. The constraints are also non-linear with respect to the plugging times. Thus, we can't use Linear Programming methods. What we meant in our reply was that the problem would be straightforward in the absence of coupling constraints (such as the aggregate consumption maximum in the case of EVs), since each agent could optimize its own cost independently. The main difficulty therefore, arises from the presence of this coupling constraint, which prevents a fully decentralized optimization.
>
> **7.** We were unable to identify a simpler example than that of electric vehicles to illustrate this control problem, as it requires the control of agents with a state space that includes at least one controllable and one uncontrollable dimension, a cost function that penalizes the use of control, and constraints on the aggregate distribution.
>
> **10.** The subsection 3.1 "Presentation of the use case" was lacking the definition of $f$ in this context; we will introduce it :
> The function $f$ is defined as: $ \forall x=(t_a,b,t_c)\in\mathcal{X},t\in[0,T]$,
>
> $$ f_t(x)=
>  p \ \text{if} \ t\in [t_c,t_c+\frac{1-b}{v}] ,$$
> $$  f_t(x)=0 \ \text{otherwise} ,$$
>
> where $p$ is the power consumption (here we normalize it to $1$) and $v=0.25 h^{-1}$ is the speed of charge of the EVs.
>
> $f_t$ is the t-th component of the vector $f$. $f^{(t)}$ is the function defined in Section 4 (end of subsection 4.1 Formulation of Online MCOT-C). In this online scenario, the constraint is defined differently at each moment of the day as the vehicles already plugged will consume a part of the maximum consumption during the rest of the day. It is therefore necessary to update this function at each time instant $t$ throughout the day, and we denote this updated version by $f^{(t)}$.
>
> **16.** We agree and will modify the contributions to highlight that we will consider the regularization of the problem.
>
> **17.** We agree and will adopt your proposed formulation.
>
> **18.** We wanted to provide an example outside of control problems to readers who are not specifically interested in control problems, and Gaussian distributions provide a standard framework for this purpose. We agree that the assumption of compactness is not met here and that the presentation is confusing; we will introduce, before Proposition 9, a remark
> to clarify it.
>
> **22.** We will use this notation.
>
> **23.** A numerical issue arises here, as our Python implementation (provided in the supplementary materials) fails when the tolerance parameter $\varepsilon$ is set to smaller values. Since there is a $\varepsilon^{-1}$ term in an exponential function in Algorithm 1, if epsilon is too small, very large numbers are obtained, which will cause Overflow for Python.
>
> **24.** This is indeed a limitation, since the control space becomes larger and, consequently, the state space $\mathcal{X}$ also increases. To clarify this limitation, we will add in the second point of the conclusion discussing how one may address higher-dimensional state spaces, that it can arise when the control becomes more complex (for instance, by allowing unplugging and replugging times).
>
> In addition, the first reviewer requested a comparison of the cost functions when this unplugging option is allowed. We have included this comparison in Appendix E (Figure 10). The conclusion is that, for this example, which is relatively easy to satisfy in terms of constraints, the two cost functions are very close.
>
> **29.** The idea is to use Monte Carlo methods to compute the gradient, which is equal to $\langle \mu_\lambda,f \rangle$ (Proposition 2). As stated in Proposition 1 (ii), $\mu_\lambda$ is proportional to $\mu_2$ with a computable coefficient $w(x,\lambda)$, thus we can generate a certain number $Z$ of $X_z\in\mathcal{X}$ with respect to $\mu_2$ and compute an approximation of the gradient $ \frac{1}{Z}\sum_{z=1}^Z f(X_z)w(X_z,\lambda)$. This modification leads to a stochastic gradient descent algorithm.

---

> ### Comment · Reviewer_Hpy6 · 2025-12-27
> **Responses to remarks 1-29**
>
> 6. I think you should highlight then how the coupling constraint renders the problem non-convex, if it does (if it is a sum over the opt variables, it is a linear constraint and the problem is a QP).
>
> 7. I meant to clearly guide the reader instantiating your notation very gradually.
>
> Please discuss explicitly the limitations in the manuscript. In particular wrt the fact that you empirically validated your method on a (small?) action space.

---

### Review · Reviewer_veQj · 2025-12-01

**Summary Of Contributions:**

The paper introduces a moment-constrained optimal transport for control (MCOT-C) framework that replaces strict second marginal constraint with more flexible moment constraints while fixing the first marginal. By incorporating a KL regularizer, the formulation yields a dual problem and enables an efficient semi-Sinkhorn algorithm. An application to the EV charging problem demonstrates the applicability of the algorithm, and justifies the study of this problem.

Strengths:
1. This paper provides a clear and flexible modeling framework, with an efficient and scalable algorithm design.
2. The problem studied in this paper is well-motivated by the EV charging application problem.

Weakness:
1. The algorithmic novelty is modest. It is a natural adaptation of Sinkhorn and gradient methods.
2. The discussion on the connections to related literatures is not adequate; see Requested Changes for details.

**Audience:**

Yes

**Audience Explanation:**

The problem formulation, algorithm design, and application in this paper would be interesting to TMLR's audience working on optimal transport, control, and their applications.

**Claims And Evidence:**

Yes

**Claims Explanation:**

The mathematical formulation, duality analysis, and algorithmic design are clearly derived and well supported. The comparisons to prior literature can be improved.

**Requested Changes:**

I recommend the authors to provide more discussions on the comparison with existing works, especially those related to moment-based OT [1] and Schrödinger Bridge [2], among others.

[1] Alfonsi, Aurélien, et al. "Approximation of optimal transport problems with marginal moments constraints." Mathematics of Computation 90.328 (2021): 689-737.
[2] Liu, Guan-Horng, et al. "Deep generalized schrödinger bridge." Advances in Neural Information Processing Systems 35 (2022): 9374-9388.

---

> ### Author Response · Authors · 2025-12-13
> **References to [1] and [2]**
>
> We thank the reviewer for these references. We will provide a more detailed comparison with [1] and add a discussion in the literature section on the second reference that we were not aware of.
>
> The main difference between MCOT-C (this work) and MCOT [1] lies in the objective and the nature of the constraints. MCOT is fundamentally an approximation of classical Optimal Transport (OT). It replaces marginal constraints with moment constraints on both marginals ($\mu_1$ and $\mu_2$) to obtain a discrete minimizer and ensure convergence to the exact OT solution. In contrast, MCOT-C is a Mean Field Control (MFC) problem. It uses moment relaxation as a modeling tool for the control objective: only the target distribution (i.e., that of the controlled agents) is subject to moment constraints, representing global system constraints such as power grid capacity. The initial distribution corresponds to the uncontrolled state. This formulation is asymmetric (unilateral), and the goal is to reach the optimal distribution (i.e., the control policy), rather than to approximate the solution of a known OT problem. In this framework, the control objectives (e.g. maximum power consumption at each time step) are not stated on the states of the agents but through the moments of their distribution.
>
> DeepGSB [2], on the other hand, focuses on Mean Field Games (MFG), a more general framework involving a Nash equilibrium among agents. It addresses a subset of MFGs where the target density $\rho_\text{target}$ is known a priori and must be reached exactly. Its methodology relies on Deep Learning and Reinforcement Learning (DeepRL) techniques. The main difference with our approach lies in the target distribution. In their case, it is known exactly, whereas we only have access to its moments. In the crowd motion examples they consider, this would correspond to knowing only the mean or the standard deviation of the crowd at the final state. Their objective is to determine the evolution from state 1 to state 2, while we focus solely on the final distribution. This final distribution, when applied to a finite number of agents, gives a control policy, which is optimal when the number of agents is very large. It would be particularly interesting to investigate, in the context of their work, what happens when the constraint on the final distribution is relaxed.

---

### Decision · Action_Editor_7NsY · 2026-01-28

**Recommendation:** Accept as is

**Additional Comments:**

The paper considers a class of a multi-agent control problems defined as minimizing a cost function while satisfying moment constraints. It is then approximated as a Mean Field Control (MFC) problem, in which the optimization is over distributions.
One of the contributions of the paper is to show that this MFC problem can be formulated as a modification of the Optimal Transport (OT) problem.
As opposed to the conventional OT problem, where both marginals are specified, here one of the marginal comes from the distribution of the non-controllable variables of the state of the agents while the other is specified by enforcing it to a set of distributions satisfying moment constraints. This is the Moment Constrained Optimal Transport (MCOT) problem.
This conversion of MFC to MCOT allowed the authors to solve the MFC problem using Sinkorn-like procedure.

Throughout the paper, the authors used an Electric Vehicle (EV) Charging problem as a motivating and illustrating example.


The three reviewers provided many comments on this paper, including the request to include another example in addition to the EV Charging problem, improvement in the related work, and many clarification requests. The authors have been engaged and answered them in detail and revised the paper accordingly.

At the end of the day, one of the reviewers's final decision is Accept, another is Leaning Accept, and the third is Leaning Reject.

The reviewer whose recommendation is Leaning Reject stated in their recommendation that they didn't "find the paper particularly novel or bringing very strong evidence. Furthermore, the exposition is not stellar".

The Leaning Accept reviewer believes that "[the] problem formulation is novel and interesting".

The other reviewer (Accept) also believes that the algorithmic novelty of this work is modest, though they found the modelling framework clear and flexible.


I believe that the Moment Constrained OT framework is interesting and novel. I also can see that the algorithm to solve it is similar to the Sinkhorn algorithm, and may not be deemed very novel.
The exposition of the paper appears to be improved from the initial submission, as I found the revised version relatively clear. This is thanks to the reviewers, especially the one who provided many detailed comments about the writing.

Since the extent of novelty is partly subjective and is not the most important criteria for TMLR, I do not believe that concerns about novelty should play the most significant role in the decision. **This is a reasonably good paper, hence I recommend its acceptance**.

There are several points where the paper can be improved, which I enlist below:

- Appendix G: The paper becomes stronger if in addition to formulating the Water Heaters problem as MCOT, the authors run some experiment too.
- During the rebuttal phase, some new appendices are added (E, F, and G), but they are not referred to in the main body of the paper. Please integrate those appendices better within the paper.
- p2: the right quotation mark in "preserving the distribution ..." does not look correct. There are a few other places where a similar issue appears.
- p4: The "but" in the sentence "This subsection defines the dual and the theoretical properties needed for the algorithm but more ..." sounds a bit strange.
- P6: The citation "C.T.Kelly, 1999" should be "Kelly, 1999".
- P12: The text close to Figure 5 is mangled.
- P13: The Water Heaters problem is mentioned in the Conclusions, but it is not mentioned that the problem is also presented in Appendix G.

**Audience:**

Yes

**Audience Explanation:**

Yes, this is relevant to researchers interested in Optimal Tranport and Control Systems.

**Claims And Evidence:**

Yes

**Claims Explanation:**

Yes, the claimed are all well-supported.

---

> ### Author Response · Authors · 2026-02-09
> **Camera-ready version**
>
> We would like to thank the reviewers for their detailed and constructive reviews, which greatly helped us improve the paper, both in terms of content and presentation. We also thank the Action Editor for the decision and have taken his comments into account to finalize the camera-ready version, which has now been submitted.
>
> In particular, regarding the first comment about the water heater example, we have provided in the appendix a link to a preprint that details the methods and the numerical experiments for the water heater control problem.